# Modular deregulation of central carbon metabolism for efficient xylose utilization in *Saccharomyces cerevisiae*

Xiaowei Li [1,2,5], Yanyan Wang[1,2,5], Xin Chen [2], Leon Eisentraut[2], Chunjun Zhan[2], Jens Nielsen [2,3] ✉ & Yun Chen [2,4] ✉

The tightly regulated central carbon metabolism in *Saccharomyces cerevisiae*, intricately linked to carbon sources utilized, poses a significant challenge to engineering efforts aimed at increasing the flux through its different pathways. Here, we present a modular deregulation strategy that enables high conversion rates of xylose through the central carbon metabolism. Specifically, employing a multifaceted approach encompassing five different engineering strategies—promoter engineering, transcription factor manipulation, biosensor construction, introduction of heterologous enzymes, and expression of mutant enzymes we engineer different modules of the central carbon metabolism at both the genetic and enzymatic levels. This leads to an enhanced conversion rate of xylose into acetyl-CoA-derived products, with 3-hydroxypropionic acid (3–HP) serving as a representative case in this study. By implementing a combination of these approaches, the developed yeast strain demonstrates a remarkable enhancement in 3–HP productivity, achieving a 4.7–fold increase when compared to our initially optimized 3–HP producing strain grown on xylose as carbon source. These results illustrate that the rational engineering of yeast central metabolism is a viable approach for boosting the metabolic flux towards acetyl–CoA–derived products on a non-glucose carbon source.

Enabling sustainable production of chemicals is the primary motivation for establishing bio-based production of fuels and chemicals. For instance, the exploitation of renewable plant-based raw materials, such as lignocellulose, for chemical production offers a more sustainable alternative to many traditional chemical processes that depend heavily on fossil fuels[1–5]. Since xylose ranks as the second most abundant sugar in lignocellulosic feedstock, its effective use is crucial in ensuring the economic feasibility of fermentation processes using lignocellulosic based feedstocks[6,7].

Another significant benefit of bio-based production of fuels and chemicals is the ability to synthesize a diverse range of biochemical molecules using a single, well-developed 'platform cell factory'. *Saccharomyces cerevisiae*, an eukaryal model microorganism and a widely used cell factory, is particularly valued due to its robustness under harsh industrial conditions and its wide use in the fermentation industry[8,9]. Recently, engineered yeast strains have successfully produced a wide array of chemically diverse molecules[10–12]. These achievements predominantly hinge on the utilization of several key metabolites in the central carbon metabolism, such as pyruvate, citric acid, α-ketoglutaric acid, and acetyl-CoA[13]. Among these, acetyl-CoA receives particular attention because it acts as a highly productive node for generating numerous biochemical products, such as

[1]Tianjin Institute of Industrial Biotechnology, Chinese Academy of Sciences, Tianjin, China. [2]Department of Life Sciences, Chalmers University of Technology, Gothenburg, Sweden. [3]BioInnovation Institute, Copenhagen N, Denmark. [4]Novo Nordisk Foundation Center for Biosustainability, Technical University of Denmark, Kongens, Lyngby, Denmark. [5]These authors contributed equally: Xiaowei Li, Yanyan Wang. ✉e-mail: nielsenj@chalmers.se; yunc@chalmers.se

isoprenoids, fatty acid derivatives, polyketides, and more[12]. Hence, developing an acetyl-CoA synthesis platform in yeast that utilizes xylose is beneficial for converting lignocellulosic biomass into value-added products.

Taken together, the effective transformation of xylose into acetyl-CoA in yeast forms a pivotal foundation for sustainable and economic feasible manufacturing of a broad range of bioproducts from lignocellulosic biomass. To achieve this, it is essential to optimize the flux through the different pathways of the central carbon metabolism. However, enhancing the flux within the central carbon metabolism presents inherent challenges due to its tightly regulated nature. These stringent regulatory mechanisms are designed to support cell growth and maintain homeostasis on the cells' favorite carbon sources, thereby potentially resisting any engineering attempts to alter this balance on the unrecognized carbon sources[1]. Previous studies on synthesizing acetyl-CoA-derived products from xylose or glucose-xylose as carbon sources focused heavily on optimizing xylose utilization and product conversion pathways to enhance the efficiency of product formation. However, it often overlooked how cells reconfigure their metabolism, including central metabolic pathways, in response to xylose utilization[7,14]. As a result, our understanding of the intracellular regulation of the central carbon metabolism during xylose utilization remains limited, which further complicates the task of

rational engineering or deregulating the central carbon metabolism to ensure high conversion rate of xylose.

In this study, we initially constructed and optimized a conversion module in yeast (Fig. 1a) that converts acetyl-CoA to 3-hydroxypropionic acid (3-HP), with the objective of directing carbon flux from xylose to acetyl-CoA and further to 3-HP. Subsequently, we divided the central carbon metabolism into three distinct modules for systematic deregulaitng the central metabolic pathways during xylose utilization (Fig. 1a). To navigate the regulatory aspects of central carbon metabolism, either at the transcriptional or enzymatic levels across different modules, we employed five distinct engineering strategies: (1) evaluating the potency of a collection of promoters in relation to xylose utilization and subsequently utilizing them for gene expression control; (2) modulating the expression of enriched transcription factors (TFs) via upregulation or downregulation; (3) introducing heterologous proteins as replacements for the extensively modified endogenous counterparts; (4) substituting the modified amino acid sites on the proteins; (5) constructing biosensors to monitor and sense intracellular metabolites levels such as NADPH and fatty acyl-CoA (Fig. 1b). By applying these strategies, our results demonstrate that despite the existence of stringent regulatory systems, it is indeed possible to manipulate the central metabolism pathway to increase flux towards desired products.

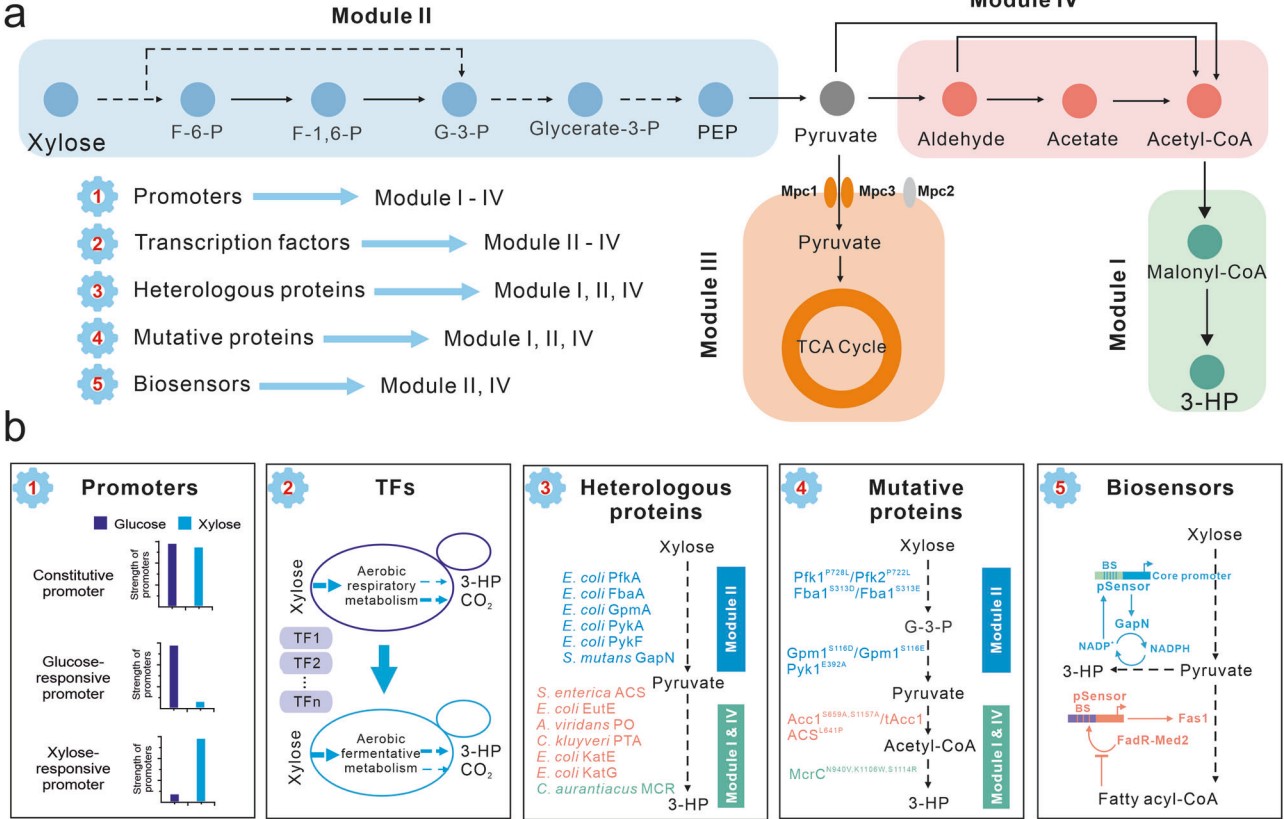

**Fig. 1 | Modular de-regulation of central carbon metabolism drives carbon flux from xylose to 3-HP. a** Schematic overview of four metabolic modules. Module I is referred to as the 3-hydroxypropionic acid (3-HP) production module. The central carbon metabolism was divided into three distinct modules: the glycolysis module (II), the mitochondrial module (III), and the PDH bypass module (IV). To deregulate each module in the central carbon metabolism, five engineering strategies were employed: (1) controlling gene expression at the transcriptional level by a series of promoters characterized on xylose; (2) manipulating global transcription factors (TFs); (3) introducing heterologous proteins; (4) expressing mutated proteins; and (5) constructing biosensors. **b** The detailed implementations of the above

mentioned five distinct engineering strategies in the various metabolism modules. BS binding sites, Enzymes and metabolites shown in the pathways: Mpc1/2/3 mitochondrial pyruvate carrier, PfkA and Pfk1/2 phosphofructokinase, FbaA and Fba1 fructose 1,6-bisphosphate aldolase, GpmA and Gpm1 phosphoglycerate mutase, PykA/F and Pyk1 pyruvate kinase, GapN NADP-dependent glyceraldehyde-3-phosphate dehydrogenase, ACS acetyl-CoA synthetase, EutE acetaldehyde dehydrogenase, PO pyruvate oxidase, PTA phosphotransacetylase, KatE/G catalase-peroxidase, Acc1 acetyl-CoA carboxylase, MCR malonyl-CoA reductase, F-6-P fructose 6-phosphate, F-1,6-P fructose 1,6-bisphosphate, G-3-P Glyceraldehyde 3-phosphate, PEP phosphoenolpyruvic acid.

## Results

### Characterizing various promoters in relation to xylose utilization

Given the crucial role of promoters in bypassing inherent regulation and driving gene expression in a desired manner, we first evaluated how a broad spectrum of promoters are controlled during xylose utilization. To achieve this, we performed an RNA-seq analysis, considering that transcript profiling can provide an effective method for assessing the cell's transcriptional response to varied carbon utilization. For this analysis, we constructed strain X266, which was transformed with a plasmid containing xylose isomerase (XI) and xylitol kinase (XK), for the conversion of xylose to xylulose, along with our previously identified modifications that ensured an efficient xylose assimilation[14] (Supplementary Data 1). Strain X266 was then cultured in xylose minimal medium, using a glucose medium as a comparison. Based on the varied gene transcriptional levels on xylose and glucose, we selected a list of promoters for evaluation, which included several that have been previously characterized and are widely used on glucose[15–17] (Supplementary Data 3). In total, we evaluated the expression of 40 different promoters during growth on either xylose or glucose, utilizing red fluorescent protein (RFP) as a reporter (Supplementary Fig. 1a).

Reflecting on the diverse RFP expression behaviors supervised by these promoters in response to xylose and glucose utilization, we categorized and defined them into three distinct groups: (1) xylose-responsive promoters, which demonstrated five times greater strength on xylose compared to glucose; (2) glucose-responsive promoters, which exhibited a reverse expression pattern relative to xylose-responsive promoters; and (3) constitutive promoters, whose expression levels showed minimal variation in gene expression during growth on xylose and glucose (Supplementary Fig. 1b–d). The strong correlation between RNA-seq data and fluorescence intensity confirms that these promoters efficiently regulate protein expression in a manner that responds to different carbon source utilization (Supplementary Fig. 1 and Supplementary Data 3). This behavior is notably different from previously characterized promoters, which were primarily derived from central metabolic pathways[18]. Next, to create a more comprehensive set of xylose-responsive promoters with a wide range of intensities, we screened an extra 13 promoters based on transcriptome data (Supplementary Data 3). We assessed the strength of each promoter by integrating them into the genome, utilizing green fluorescent protein (GFP) as a reporter due to its superior fluorescence intensity, thereby providing more suitable evaluation of weaker promoters[19]. By culturing the constructed strains in xylose minimal medium, we observed that the measured promoters spanned a 42-fold range of GFP fluorescence intensities, while these promoters exhibited relatively low fluorescence intensities in glucose minimal medium (Fig. 2a).

Natural systems tend to favor regulons, or nutrient-responsive gene regulations that manage numerous cellular functions, such as nutrient-assimilation pathways, which are critical for cell survival and growth. For example, previous research has shown that regulon-controlled expression of galactose catabolic genes supports higher growth rates compared to their constitutive expression[20]. To explore whether employing xylose-responsive promoters for controlling the expression of catabolic genes could likewise boost xylose utilization efficiency, we substituted the promoters of XI, a pivotal step in controlling xylose assimilation[14], with *pADH2* and *pSFC1*, two xylose-responsive promoters with strong and moderate strength, respectively. Upon assessing cell growth, we found that *pADH2*-controlled XI resulted in faster cellular growth compared to the use of the constitutive promoter *pTEF1*, thereby demonstrating the advantage of using xylose-responsive promoters (Fig. 2b).

### Optimizing the 3-HP bioconversion module

Next, to reflect the strength of the metabolic flux from xylose through central carbon metabolism to acetyl-CoA, it is crucial to channel this intermediate towards a designated product. While the choice of such a product must (1) circumvent the introduction of highly regulated metabolic pathways, such as fatty acid or terpene biosynthesis, and (2) avoid the requirement for several intermediate metabolites, such as flavonoid synthesis (Supplementary Fig. 2). We therefore introduced a conversion module (Module I) to produce 3-HP (Fig. 1a), that was chosen due to its straightforward synthetic pathway from acetyl-CoA/malonyl-CoA.

In Module I, we reconstructed a 3-HP biosynthesis pathway by introducing the bifunctional enzyme MCR from *Chloroflexus aurantiacus*, which catalyzes the conversion of malonyl-CoA to 3-HP in two steps[21,22], using malonate semialdehyde as an intermediate (Supplementary Fig. 3a). The incorporation of MCR in a plasmid resulted in the production of 74 mg/L of 3-HP from 20 g/L of xylose (Supplementary Fig. 3b). Next, to optimize Module I, we utilized glucose as the carbon source. This simplified the genetic manipulation steps by eliminating the need to consider xylose metabolism and also provided an opportunity to compare the performance of these two carbon sources. Previous research has indicated that splitting MCR into McrN and McrC components could improve enzyme activity, while a mutated form of McrC (McrCm) further boosted 3-HP production in *Escherichia coli*[23]. Applying these approaches, we observed consistent outcomes across the engineered yeast strains, resulting in a 3-HP level of up to 1.2 g/L (Supplementary Fig. 3c). However, this splitting strategy challenges the commonly observed benefit of protein fusion strategy to facilitate substrate trafficking[24,25]. To investigate this anomaly, we aggregated McrN and McrCm into three different fusion forms, including the wild-type form (McrN-Cm) and two linker-based fusions (McrN-linker-McrCm and McrCm-linker-McrN). However, none of these constructions increased 3-HP production, suggesting that physical proximity does not necessarily facilitate substrate trafficking between these two catalytic components (Supplementary Fig. 3c). Moving forward, to create a stable conversion module capable of incorporating generated acetyl-CoA, we integrated the McrN-Cm gene into the chromosome XI-3 locus. Unexpectedly, reducing the copy number of McrN-Cm led to a significant decrease in the yield of 3-HP, falling to 116 mg/L from 20 g/L of glucose. Subsequently, we fine-tuned the ratio of these two components by controlling McrN with 5 different promoters of varying strengths. This strategy successfully increased the yield of 3-HP to 2.1 g/L when McrN was regulated by the *pENO2* promoter and McrCm was controlled by the *pHXT7* promoter (Supplementary Fig. 4), which emphasized the balance between these two catalyzing components.

When the xylose metabolic pathway was introduced into the best producing strain on glucose, there was a dramatic decrease in 3-HP production in xylose medium (Fig. 2c). Interestingly, when we substituted the McrN promoter with *pTEF1*, we observed a 1.2-fold increase in 3-HP levels, a pattern distinct from that observed on glucose (Fig. 2c and Supplementary Fig. 4). Next, given the demonstrated benefits of the xylose-responsive promoter in enhancing xylose utilization (Fig. 2b), we replaced the promoter of McrN with *pALD4* (strain R30c), resulting in a 92% improvement in 3-HP levels compared to those achieved with the McrN promoter using *pTEF1* (Fig. 2c). With this optimized 3-HP converting module on xylose, we proceeded to improve the metabolic flux through the central carbon pathway from xylose.

### Engineering global TFs to enhance the metabolic flux through central carbon metabolism

Control over the flux through central carbon metabolism requires amplifying the activities of numerous enzymes within this pathway to achieve an increase in flux. A potential strategy to bypass this issue could involve the engineering of regulatory networks that govern central carbon metabolism. It is important to note that the cell's central carbon metabolism varies across different carbon sources through

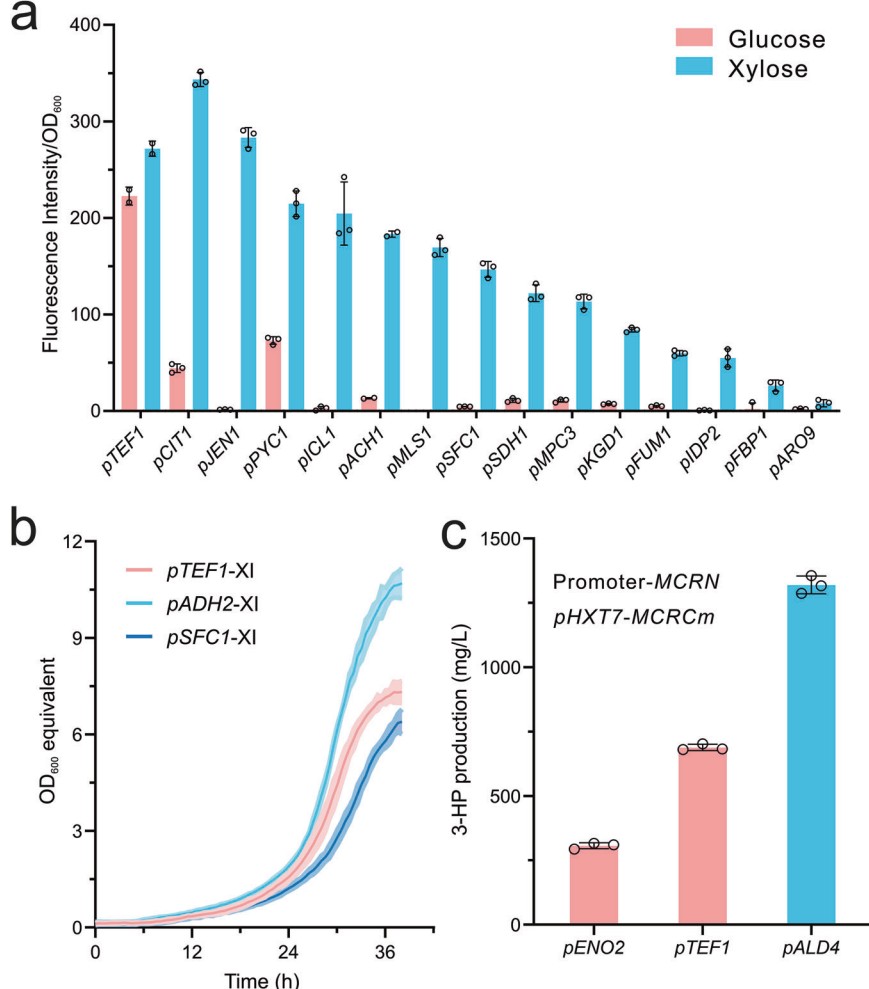

**Fig. 2 | Characterization and application of xylose-responsive promoters.**
**a** Characterization of a set of xylose-responsive promoters on both glucose and xylose. The constitutive promoter *pTEF1* was used as an internal standard. The green fluorescent protein (GFP) was used as a reporter. **b** Growth curves of three engineered xylose metabolizing strains. Two xylose-responsive promoters (*pADH2* and *pSFC1*), as well as a constitutive promoter (*pTEF1*), were used to control the expression of XI, an initial and key enzyme for xylose metabolism. **c** Evaluating the conversion efficiency of malonyl-CoA to 3-HP. Two constitutive promoters (*pENO2* and *pTEF1*) and a xylose-responsive promoter (*pALD4*) were used to control the

expression of *MCRN*. Promoter *pHXT7* was used to control the expression of *MCRCm*. For (**a**), strains were cultivated in a BioLector using a minimal medium with 2% glucose or xylose as the carbon source. For (**b**), strains were cultivated in a Growth Profiler using a minimal medium with 2% xylose as the carbon source. For (**c**), strains were cultivated in shake flasks using a minimal medium with 2% xylose as the carbon source. All data represent the mean derived from *n* = 3 biologically independent samples, with error bars indicating the standard deviation. Source data are provided as a Source Data file.

changing the expression of specific TFs[26]. Numerous studies have demonstrated that when *S. cerevisiae* strains metabolize xylose, they perceive it as a non-fermentable carbon source, consequently triggering an aerobic respiratory metabolism[27–29]. However, the metabolic flux that transforms xylose into acetyl-CoA through glycolysis and the pyruvate branching pathway is close to the yeast's fermentation metabolism on glucose (Supplementary Fig. 2). Therefore, to increase the metabolic flux from xylose to acetyl coenzyme A, we propose engineering TFs to simulate the metabolic state of central carbon metabolism typically observed when glucose is the carbon source.

To identify TFs that govern glucose metabolism, we undertook a differential TF analysis. During this analysis, we considered two sets of input data: (1) the entire genomic catalog of yeast genes, and (2) a subset of genes crucially linked to central metabolism (Fig. 3a). This subset included genes related to glycolysis, gluconeogenesis, the tricarboxylic acid (TCA) cycle, the pentose phosphate pathway (PPP), oxidative phosphorylation, respiration metabolism, pyruvate metabolism, the glyoxylate cycle, as well as carbon transport and metabolism. By

integrating the transcriptional differences of genes under glucose and xylose conditions, along with their regulatory network interactions with transcription factors, we scored 150 and 125 TFs, respectively (Supplementary Data 6 and 7). The top 10 TFs that were significantly up-regulated and the top 10 TFs that were significantly down-regulated are presented in Fig. 3b. Interestingly, half of the enriched TFs were shared for both inputs, implying that the regulatory shifts in yeast cells on glucose and xylose mainly involve central carbon metabolism. Next, we checked the changes in these TFs' expression levels on glucose and xylose. We then selected those TFs whose observed shifts in expression levels aligned with the up- and down-regulation patterns identified from our reporter TF analysis for overexpression or deletion (Fig. 3b). Ultimately, we tested 18 TFs and found that overexpressing *MBP1*, *GCR1*, and *GCR2* enhanced the flux towards 3-HP production from 11% to 22% compared to the control strain (Fig. 3c). Meanwhile, deleting *SIP4* and *Mig1* increased the flux through the central pathway by 9% and 11%, respectively (Fig. 3d). Notably, the removal of genes related to the HAP complex, a significant regulator of respiratory gene expression, severely reduced the flux towards 3-HP production as well as cellular growth

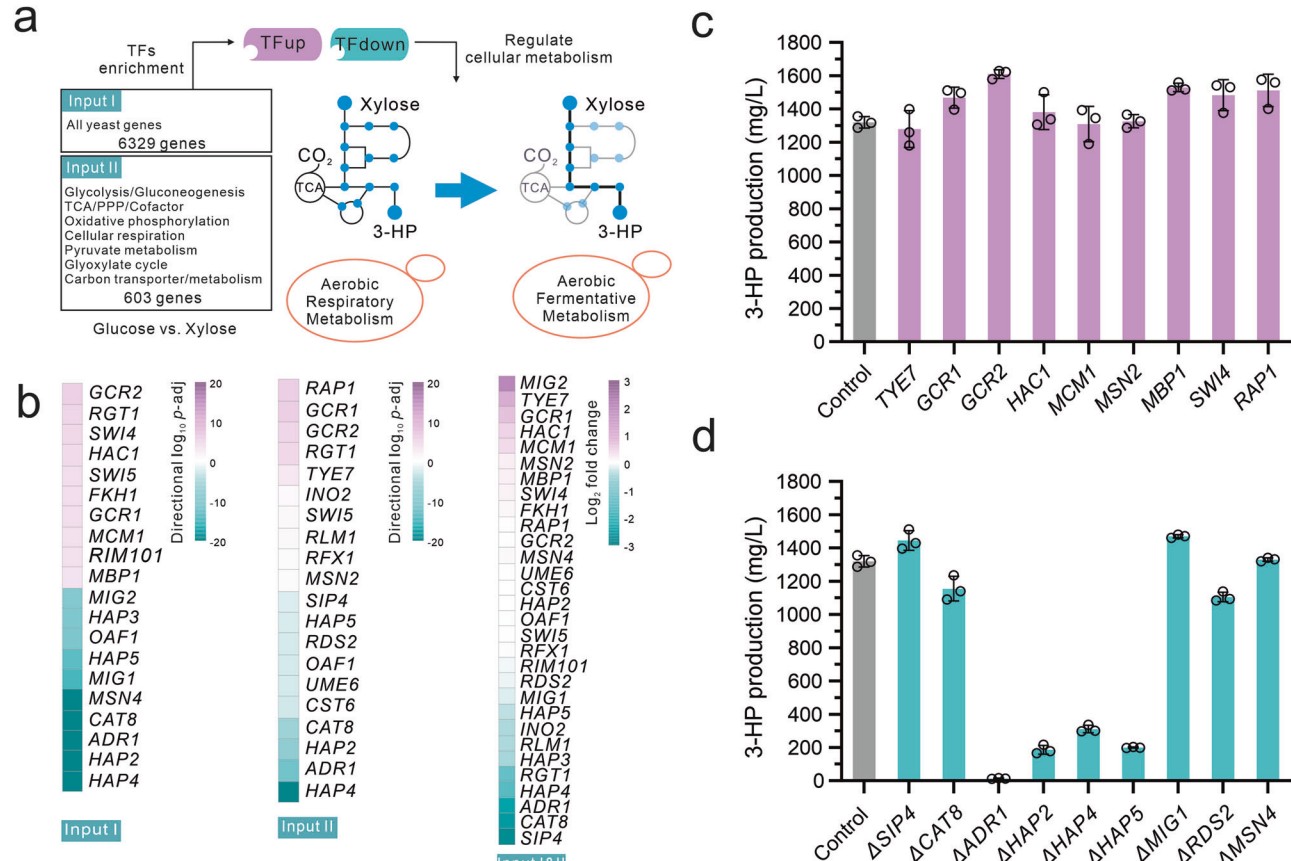

**Fig. 3 | Engineering global TFs to increase metabolic flux through central carbon metabolism. a** Schematic illustrating TF analysis. To shift from 'Aerobic Respiratory Metabolism' to 'Aerobic Fermentative Metabolism' when using xylose, a TF analysis was conducted. This analysis utilized two sets of input data: the comprehensive yeast genome, encompassing 6329 genes (input I), and a subset of 603 genes related to central metabolism (input II). **b** The top 10 enriched TFs in both the upregulated and downregulated classes were identified using either gene input I or input II. Expression levels of the 30 overlapping TFs from inputs I and II were documented. *p*-adj (adjusted *p*-values, calculated by the Benjamini–Hochberg method). **c** Experimental validation of the significantly upregulated TFs via gene overexpression. **d** Experimental validation of the significantly downregulated TFs via gene deletion. For (**c**) and (**d**), strain R30c was used as the control. Strains were grown in a minimal medium with 2% xylose as the sole carbon source. For (**c**) and (**d**), all data represent the mean derived from *n* = 3 biologically independent samples, with error bars indicating the standard deviation. Source data are provided as a Source Data file.

(Fig. 3d and Supplementary Fig. 5b), suggesting the necessity for meticulous regulation of cellular respiration.

Further investigation revealed that the combined overexpression of *MBP1*, *GCR1*, and *GCR2* enhanced 3-HP production by 29% (Supplementary Fig. 6). This moderate boost in 3-HP production suggests that improving the flux through engineering of a set of TFs is a challenging task. Given glucose's significant advantage over xylose in synthesizing acetyl-CoA through central metabolism, this benefit is fundamentally attributed to two crucial regulatory mechanisms present under glucose conditions: the carbon catabolite repression (CCR) effect[30] and the Crabtree effect[31]. These mechanisms, which are less pronounced under xylose conditions, primarily involve glycolysis, mitochondrial metabolism, and pyruvate dehydrogenase (PDH) bypass. Consequently, we have divided the metabolic pathway from xylose to acetyl-CoA into three distinct modules: the glycolysis module, the mitochondrial module, and the PDH bypass module (Fig. 1a). Our next step was to systematically deregulate each module, thereby promoting metabolic flux through the central pathway and directing the flux toward 3-HP production.

## Deregulation of glycolysis module

Previous studies demonstrated that, in contrast to glucose, xylose does not fully activate the CCR response in *S. cerevisiae*, resulting in diminished carbon flux through the glycolytic pathway[29,32]. However, optimizing the glycolytic pathway poses a formidable challenge, as it represents one of the most complex and tightly regulated routes within the central metabolic pathways[33]. To mitigate this regulatory structure and increase the flux through glycolysis, we executed the following major modifications: (1) blocking gluconeogenesis and the glyoxylate cycle, (2) increasing the expression levels of glycolytic enzymes, (3) integrating glycolytic enzymes from E. coli, (4) expressing key mutated enzymes in glycolysis, and (5) engineering effector and cofactor metabolism.

Considering the differential CCR response on glucose and xylose, we first analyzed the transcriptional levels of central carbon metabolism-related genes under both conditions (Supplementary Fig. 7). Our results indicated that when xylose was the carbon source, genes involved in glycolysis were downregulated, while those involved in gluconeogenesis and the glyoxylate cycle were upregulated, compared to their expression on glucose. Additionally, we examined the transcriptional levels when using ethanol, a fully gluconeogenic carbon source requiring flux through the reverse glycolytic direction. We found that, compared to their expression on ethanol, the genes involved in glycolysis were upregulated, and those involved in gluconeogenesis and the glyoxylate cycle were downregulated. These findings suggest that when cells grow on xylose, enzymes from both the gluconeogenic and glycolytic pathways are active and operate in opposite directions,

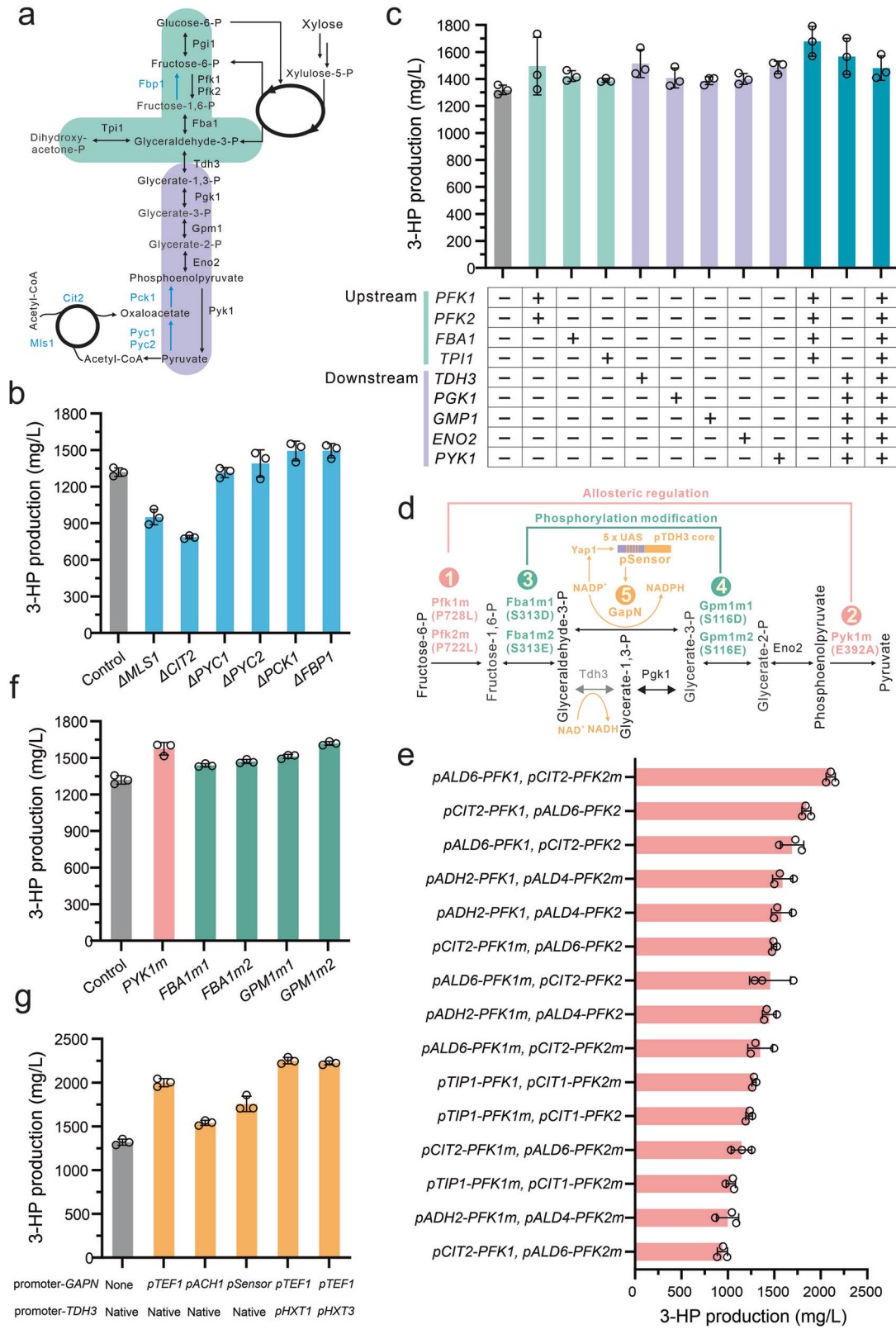

potentially leading to futile cycling (Fig. 4a). To validate this, we proceeded to delete 6 genes involved in the gluconeogenesis pathway and the glyoxylate cycle (Fig. 4a). The deletion of *PCK1* and *FBP1*, which are involved in the gluconeogenesis pathway, led to a significant increase in 3-HP production ($P < 0.05$). In contrast, deleting *MLS1* and *CIT2*, which are associated with the glyoxylate cycle, resulted in a decrease in 3-HP production (Fig. 4b). This

suggests that while the gluconeogenesis pathway contributed somewhat to futile cycling, the glyoxylate cycle did not.

Next, to identify the pivotal steps that significantly influence glycolytic flux, we initially assessed the expression levels of glycolytic enzymes. As illustrated in Fig. 4a, we divided the linear glycolytic pathway into two segments: 'upstream glycolysis', which spans from fructose-6-phosphate to glyceraldehyde-3-phosphate, and

**Fig. 4 | Deregulation of glycolysis module. a** Schematic illustration of the gluconeogenesis and glycolysis pathways. The gluconeogenesis pathway is marked in blue, while the linear glycolytic pathway is divided into 'upstream glycolysis' (green) and 'downstream glycolysis' (purple). **b** Deletion of 6 individual genes within the gluconeogenesis pathway. **c** Single and combinatorial overexpression of 9 genes within the glycolysis pathway. **d** Schematic depicting the mutated enzyme within the glycolysis pathway. Two allosterically regulated enzymes, Pfk1/2 and Pyk1, are highlighted in pink, while two phosphorylation-modified enzymes, Fba1 and Gpm1, are shown in green. The NADPH biosensor is marked yellow. **e** Regulation of Pfk1 and Pfk2, or their mutant versions, affects the flux through glycolysis. **f** Overexpression of mutated Pyk1, Fba1, and Gpm1. **g** Engineering of the cofactor metabolism for NADPH and NADH within the glycolysis pathway. Enzymes shown in the pathways: Mls1 malate synthase, Cit2 citrate synthase, Pyc1/2 pyruvate carboxylase, Pck1 phosphoenolpyruvate carboxykinase, Fbp1 fructose-1,6-bisphosphatase, Pgi1 phosphoglucose isomerase, Tdh3 glyceraldehyde-3-phosphate dehydrogenase, Pgk1 3-phosphoglycerate kinase, Eno2 phosphopyruvate hydratase. See Fig. 1 and its accompanying legend for details on other enzyme information. For (**b**, **c**) and (**f**, **g**), strain R30c was used as the control. All strains were cultivated in a minimal medium containing 2% xylose as the sole carbon source. All data presented in this figure represent the mean from $n = 3$ biologically independent samples, with error bars indicating the standard deviation. Source data are provided as a Source Data file.

'downstream glycolysis', which extends from glyceraldehyde-3-phosphate to pyruvate. We then evaluated the impact of each step on regulating metabolic flux through glycolysis by overexpressing the corresponding gene for each step in the pathway. The overexpression of these genes resulted in a modest enhancement of 3-HP yield, with the most significant improvement being a 13% increase observed by the overexpression of *PYK1* (Fig. 4c). Additionally, the combinational overexpression of 4 genes (*PFK1*, *PFK2*, *FBA1*, and *TPI1*) from 'upstream glycolysis' resulted in a 27% improvement in 3-HP production, while the simultaneous overexpression of five genes (*TDH3*, *PGK1*, *GMP1*, *ENO2*, and *PYK1*) from 'downstream glycolysis' led to a 19% increase. Interestingly, the co-overexpression of all nine genes surprisingly resulted in a decrease in 3-HP synthesis compared with overexpressing either the upper or lower glycolytic enzymes alone, indicating that a balanced interplay between the upstream and downstream glycolysis is crucial (Fig. 4c).

Previous research has shown that in yeast, thermodynamically favored enzymes, such as phosphofructokinase (Pfk1 and Pfk2), fructose bisphosphatase (Fba1), phosphoglycerate mutase (Gpm1), and pyruvate kinase (Pyk1), undergo extensive post-translational modifications in yeast[34]. These modifications play an important role in modulating their enzymatic activities and, consequently, the glycolytic flux. To circumnavigate the complications arising from the regulation of endogenous enzymes, we opted to introduce enzymes from *E. coli* into yeast. The introduction of *fbaA*, *gpmA*, and *pykA* enhanced 3-HP production by 12%, 10%, and 19%, respectively. In contrast, the expression of *pfkA* resulted in a 25% reduction in 3-HP production, demonstrating a different regulation pattern of PFK in yeast (Supplementary Fig. 8).

An alternative approach to avoid endogenous enzyme modifications is to express enzymes with mutated regulation sites. To further explore the key regulatory steps within glycolysis, we focused on investigating the effects of modifications on these four enzymes. Among them, Pfk1 and Pyk1 are subject to allosteric regulation, while Fba1 and Gpm1 undergo phosphorylation modifications (Fig. 4d)[34–37]. A previous study demonstrated that a mutation in the enzyme Pfk1 (P728L) renders the enzyme capable of allosteric activation without requiring AMP or 2,6-bisphosphofructose[37]. The introduction of the P728L mutation into Pfk1 and/or its potential regulatory counterpart Pfk2 (P722L) (Supplementary Fig. 9) led to a decrease in 3-HP production compared to their respective wild-type versions (Fig. 4c, e). This unexpected result prompted us to hypothesize that the mutant site might have caused a disparity in the activities of the Pfk1 and Pfk2 subunits, thereby reducing the activity of the resulting heterotetramer. To test this hypothesis, we experimented with both promoter substitutions and mutation introductions. This approach resulted in 3-HP production varying from 0.9 g/L to 2.1 g/L, a 2.2-fold disparity (Fig. 4e), indicating that the PFK step serves as a crucial control point within the glycolytic pathway. Similarly, we overexpressed a mutated version of Pyk1 (E392A), which eliminated the cooperativity and allosteric regulation[36], resulting in a 19% enhancement of 3-HP production (Fig. 4f). To further investigate whether the expression level of the mutant Pyk1 has an impact on glycolytic flux, we changed its promoter from *pPCK1* to *pTPI1*. This switch did not obviously alter 3-HP production (Fig. 4f and Supplementary Fig. 10), demonstrating that, unlike PFK, PYK is not a major regulatory point. In addition to allosteric regulation, phosphorylation modification serves as another vital and rapid form of metabolic regulation. Within the glycolytic pathway, Fba1 and Gpm1 are two enzymes that undergo a high degree of phosphorylation[34], being modified at a singular locus. To explore whether negating the phosphorylation modification through point mutations on these two enzymes would substantially influence glycolytic flux, we proceeded to overexpress their mutated variants. Specifically, we substituted the modified serine residue with either aspartic acid or glutamic acid, consequently creating Fba1m1 (S313D), Fba1m2 (S313E), Gpm1m1 (S116D), and Gpm1m2 (S116E) (Fig. 4d). The two Fba1 mutant variants demonstrated comparable 3-HP synthesis yields to those observed when overexpressing the wild-type Fba1. Conversely, the two Gpm1 mutant variants resulted in a higher yield of 3-HP compared to overexpressing the wild-type Gpm1, with Gpm1m2 showing a 17% increase in yield relative to the wild type (Fig. 4c, f). These results suggest that phosphorylation acts as a negative regulator of Gpm1 in controlling its activity.

Beyond manipulating enzymes directly involved in glycolytic pathway, we also examined the impact of effectors, such as fructose-2,6-bisphosphate, and cofactors, including ATP and NADPH, on 3-HP synthesis. As fructose-2,6-bisphosphate is a powerful positive allosteric effector that activates glycolysis[38], we overexpressed the key gene *PFK27* using a range of promoters with varying strengths, aiming to increase its levels while avoiding excessive amounts that could cause carbon loss. The results showed that, under the control of promoter *pARO9*, the production of 3-HP improved by 15% (Supplementary Fig. 11). Given that a reduction in intracellular ATP levels can boost glycolytic flux, we opted to add benzoate, a non-metabolizable weak acid whose transport can result in decreased cellular ATP levels (Supplementary Fig. 11a), to the xylose medium[39]. While the overall level of 3-HP did not increase with the addition of benzoic acid, the amount per unit of $OD_{600}$ biomass was significantly elevated (Supplementary Fig. 11b, c), suggesting a potential increase in flux through the glycolytic enzymes. This was further evidenced by the production of 0.8 g/L acetate when 16 mM benzoic acid was added (Supplementary Fig. 11d). NADH levels are another crucial factor influencing glycolytic flux[40]. However, since NADPH plays a significant role in the synthesis of 3-HP, we aimed to strike a balance between reducing NADH levels while simultaneously increasing NADPH levels to optimize the production of 3-HP. This was achieved by introducing a NADP+-dependent glyceraldehyde 3-phosphate dehydrogenase (GapN)[41], which was regulated using either the *pTEF1* and *pACH1* promoters or a constructed biosensor promoter designed to activate GapN expression when NADPH levels are low[42]. Interestingly, utilizing the *pTEF1* promoter to control GapN expression resulted in a maximum increase of 3-HP synthesis, with a 52% improvement compared to the control R30c (Fig. 4g). Additionally, substituting the native promoter of Tdh3, an enzyme responsible for generating intracellular NADH, with the glucose-responsive promoter *pHXT1*, further increased the production of 3-HP by 13% (Fig. 4g).

## Engineering pyruvate transporters to control carbon flux entering mitochondria

A notable characteristic of *S. cerevisiae* metabolism under aerobic conditions with an excess of glucose is the Crabtree effect, which diverts the carbon flux of pyruvate towards ethanol production[31]. This effect is often associated with low respiratory efficiency, a phenomenon attributed to glucose inhibition. When xylose was used as a carbon source, genes associated with ethanol synthesis were significantly downregulated, while those for ethanol consumption were upregulated, in comparison with glucose (Supplementary Fig. 13). Additionally, genes related to the TCA cycle and respiration were upregulated (Supplementary Fig. 14). These observations suggest that xylose does not exhibit a significant Crabtree effect. Instead, the increased activity of the TCA cycle and respiratory metabolism leads to carbon loss. Thus, to minimize the diversion of carbon into the mitochondria and instead direct the flux toward acetyl-CoA synthesis, we aimed to engineer the pyruvate transporter, a crucial gatekeeper for mitochondrial entry.

In *S. cerevisiae*, the mitochondrial pyruvate carrier (Mpc) facilitates the transport of cytoplasmic pyruvate into the mitochondria and is encoded by three genes: *MPC1*, *MPC2*, and *MPC3*[43]. These genes form two distinct transportation complexes: Mpc1-Mpc2 and Mpc1-Mpc3, referred to as $MPC_{FERM}$ and $MPC_{OX}$, respectively, with the latter exhibiting greater transport activity[44]. Concurrently, yeast cells selectively express these two complexes based on nutrient availability. Therefore, we first investigated the expression levels of these three genes in the presence of xylose. We observed a substantial decrease in *MPC2* transcripts and a significant increase in *MPC3* transcripts compared to glucose (Fig. 5a and Supplementary Fig. 13). Conversely, the transcript levels of *MPC2* and *MPC3* showed opposite trends when compared to ethanol (Supplementary Fig. 13). To determine the impact of *MPC2* on pyruvate transport, we employed a glucose-responsive promoter to regulate its expression but found no notable difference in 3-HP levels, indicating that *MPC2* doesn't play a pivotal role in this process (Fig. 5b). Subsequently, we fine-tuned the expression of *MPC3* by utilizing a series of xylose-responsive promoters with relatively weaker strength compared to *pMPC3* (Fig. 2a), aiming to regulate mitochondrial pyruvate import. By employing the promoter *pARO9* to regulate *MPC3*, we successfully channeled more carbon flux towards acetyl-CoA formation, resulting in a 29% enhancement of 3-HP production relative to the parental strain R30c (Fig. 5b).

## Redirecting carbon flux toward PDH bypass pathway

Considering attenuation of the Crabtree effect during xylose metabolism, the conversion of pyruvate into acetyl-CoA may be less efficient. In fact, the PDH bypass pathway is tightly regulated in yeast, thereby resulting in a controlled flux through this crucial metabolic route[45]. To tackle this issue, we employed various engineering strategies, including the overexpression or mutation of the regulated enzymes, introduction of exogenous pathways, and construction of new regulatable systems by introducing biosensors. These approaches were designed to (1) redirect the flux towards acetyl-CoA formation, (2) further drive the flux towards malonyl-CoA formation, and (3) autoregulate fatty acid accumulation.

In yeast, pyruvate is converted to acetyl-CoA in cytosol by three enzymes: pyruvate decarboxylase (PDC), aldehyde dehydrogenase (ALD), and acetyl-CoA synthetase (ACS) (Fig. 5c). Among them, only the transcription level of *PDC1* was relatively low when xylose was used as a carbon source, compared to glucose (Supplementary Fig. 13). Therefore, we overexpressed the *PDC1* gene under the control of promoters *pCIT2* and *pMDH2*. The results indicated that using the *pCIT2* promoter led to a slight improvement in 3-HP production, with a 9% increase compared to the parental strain R30c (Fig. 5d), suggesting that PDC is not a main limiting step during this conversion. Furthermore, given that ACS is strictly regulated in yeast through acetylation,

we introduced a variant of ACS (L641P) from Salmonella enterica to avoid regulatory inhibition by acetylation[46]. However, unexpectedly, overexpressing ACS (L641P) using the *pCCW12* promoter led to a reduction in 3-HP synthesis, highlighting the unpredictability of ACS in controlling acetyl-CoA production (Fig. 5d). To address this issue, we introduced the enzyme EutE from *E. coli*, which can directly convert acetaldehyde to acetyl-CoA[47], resulting in a 40% improvement in 3-HP production compared to strain R30c. Next, to circumvent the native regulatory mechanisms of the complete PDH bypass pathway, we introduced pyruvate oxidase (PO) from *Aerococcus viridans* and phosphotransacetylase (PTA) from *Salmonella enterica* to enable the direct conversion of pyruvate to acetyl-CoA[48], utilizing acetyl phosphate as an intermediate (Fig. 5c). As acetyl phosphate can be converted to acetate by the native glycerol-3-phosphate phosphatase (Gpp1)[47], we subsequently deleted the GPP1 gene in the parental strain R30c, creating strain R70. Unexpectedly, the expression of PO and PTA in R70 resulted in a sharp decrease in 3-HP production and a reduced biomass formation (Fig. 5e and Supplementary Fig. 15c). We speculated that the generation of $H_2O_2$ during the catalysis process could be the potential reason. However, the overexpression of either native or exogenous catalases, which play a role in $H_2O_2$ detoxification, failed to restore the biomass and 3-HP levels (Supplementary Fig. 15b, c). This result led us to hypothesize that additional types of oxide molecules might have been generated in addition to $H_2O_2$. To investigate this possibility, we overexpressed several other antioxidant genes[49], namely *AHP1*, *CCP1*, *YAP1*, *SOD1*, and *SOD2*. Of the genes tested, the simultaneous overexpression of *SOD1* and *SOD2*, which are responsible for detoxifying superoxide, restored biomass formation and resulted in a notable enhancement of 3-HP production, with a 34% improvement observed compared to strain R70 (Fig. 5e and Supplementary Fig. 15b, c).

The conversion of acetyl-CoA to malonyl-CoA is catalyzed by acetyl-CoA carboxylase Acc1, the transcription of which is meticulously regulated by TFs, and the activity of which is primarily controlled through phosphorylation[50]. Previous attempts to eliminate phosphorylation regulation by substituting two serine residues (S659 and S1157) in Acc1 with alanine led to a substantial increase in fatty acid production[51]. To deregulate Acc1 and optimize its expression level when grown on xylose, we expressed the modified Acc1 (S659A and S1157A) using various xylose-responsive promoters. Our results demonstrated that utilizing promoter *pKGD1* resulted in the most notable increase in 3-HP production, showing a 1.1-fold enhancement compared to the parental strain R30c (Fig. 5f). Although the point-mutated Acc1 led to a vast improvement, there was a reported risk of back mutations to its wild-type form. Consequently, we attempted an alternative approach, in which the regulatory loop of Acc1, spanning residues S1137 to S1170, was substituted with a flexible linker (GGGG) to prevent phosphorylation modifications[52]. This approach resulted in an 88% elevation in 3-HP production compared to the R30c strain.

As a crucial connector between central carbon metabolism and fatty acid synthesis, fatty acid synthase, comprised of Fas1 and Fas2, is stringently regulated within yeast cells. Fas1 and Fas2 undergo co-regulation at the transcriptional level by general transcription factors including Reb1, Rap1, and Abf1, as well as the phospholipid-specific heterodimeric activator Ino2/Ino4[53,54]. Additionally, the expression of Fas2 is positively autoregulated by the level of Fas1[55]. To navigate the native complex regulation and minimize the overflow of carbon flux into fatty acid synthesis, we built a regulatable system that utilizes biosensors to detect cellular levels of fatty acyl-CoA (Fig. 5g). In this biosensor system, we fused the Gal4 activation domain and the yeast transcriptional mediator Med2 separately with FadR, which is responsive to cellular acyl-CoA levels[56], thereby creating two synthetic activators (Fig. 5g). The expression of these activators was regulated by either the promoter *pTEF1* or *UAS-pTDH3*, aiming to fine-tune their expression levels. Additionally, we inserted 1 or 3 binding sites (BSs) for

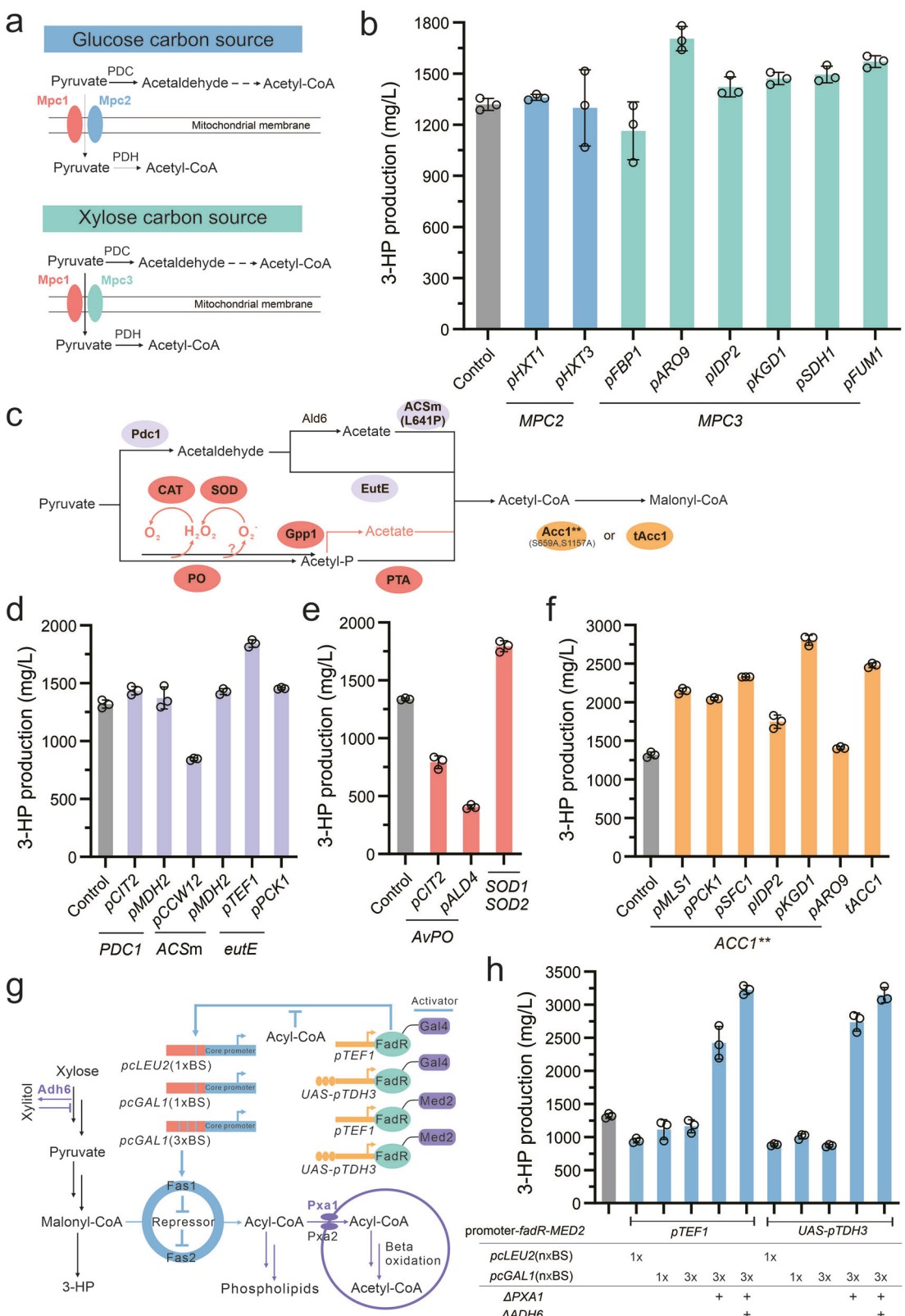

FadR into the upstream region of the *pLEU2* and *pGAL1* core promoters. These modified promoters were subsequently utilized to control the expression of Fas1. Through this design, the fused FadR activator binds to the BSs in the modified promoters to activate Fas1 expression when acyl-CoA levels are low. Conversely, as the levels of acyl-CoA increase, the activator dissociates from the promoter, deactivating transcription and consequently downregulating Fas1

expression. Unexpectedly, integrating this biosensor system resulted in lower 3-HP production (Fig. 5h and Supplementary Fig. 16a), likely due to the low levels of acyl-CoA accumulation. In response, we knocked out the *PXA1* gene, which is essential for importing acyl-CoA into peroxisomes. This modification significantly increased 3-HP production, with the most notable improvements observed when the FadR-Med2 activator was expressed in conjunction with the modified

**Fig. 5 | Reprogramming yeast cellular carbon metabolism through pyruvate node. a** Schematic overview of the proposed polymerized forms of pyruvate transport proteins. Mpc1 and Mpc2 form the complete transporter when glucose is used as the carbon source. Whereas, when xylose is used, Mpc1 and Mpc3 combine to form the complete version. **b** Fine-tuning the expression of *MPC2* and *MPC3* using either glucose-responsive or xylose-responsive promoters. **c** Schematic overview of different metabolic pathways from pyruvate to malonyl-CoA. **d** Comparison of two distinct pyruvate-to-acetyl-CoA pathways that use acetaldehyde as an intermediate. **e** Introduction of a heterologous pyruvate-to-acetyl-CoA pathway that utilizes acetyl-phosphate as the intermediate. **f** Deregulation of acetyl-CoA carboxylase. **g** Schematic representation of fatty acyl-CoA biosensor construction. The acyl-CoA binding protein, FadR, was fused with either the transcriptional activator Gal4 or Med2. Either one or three binding sites (BS) for FadR were incorporated into the upstream region of the *pLEU2* or *pGAL1* core promoters.

These engineered promoters then replaced the native promoter of Fas1. **h** Evaluation of 3-HP production with the constructed acyl-CoA biosensor. Enzymes shown in the pathways: Pdc1 pyruvate decarboxylase, Ald6 aldehyde dehydrogenase, ACSm acetyl-CoA synthetase mutant (L641P), CAT catalase-peroxidase, SOD superoxide dismutase, Gpp1 glycerol-3-phosphate phosphatase, Acc1** Acetyl-CoA carboxylase double mutant (S659A, L1157A), tAcc1 truncated Acc1, Ald6 aldehyde dehydrogenase, Pxa1 acyl-CoA transport. See Fig. 1 and its accompanying legend for details on other enzyme information. For (**b**), (**d**), (**f**), and (**h**), strain R30c was used as the control. For (**e**), strain R70, derived from strain R30c by deleting *GPP1*, was used as a control strain. All strains were cultivated in a minimal medium containing 2% xylose as the sole carbon source. All data presented in this figure represent the mean from 3 biologically independent samples, with error bars indicating the standard deviation. Source data are provided as a Source Data file.

*pGAL1* promoter containing three binding sites (Fig. 5h and Supplementary Fig. 16a). Notably, we observed a sharp increase in xylitol levels in these engineered biosensor strains (Supplementary Fig. 16a, e). Our previous research uncovered a connection between the expression of Adh6 and fatty acid metabolism, with Adh6 playing a pivotal role in xylitol formation. This finding was further corroborated by the increase in xylitol levels when we downregulated the expression of Fas1 (Supplementary Fig. 16b, c). Following the deletion of Adh6 from the engineered biosensor strain, the xylitol level decreased, and the 3-HP level reached 3.1 g/L, representing a 1.4-fold enhancement compared to strain R30c (Fig. 5h and Supplementary Fig. 16e).

## Combination of the strategies to enhance the flux toward 3-HP and flaviolin formation

After investigating the deregulated steps within the central metabolic pathway, we subsequently integrated these deregulation strategies with the goal of channeling more carbon flux towards 3-HP production (Fig. 6a). Initially, by deleting the *FBP1* and *PCK1* genes from the gluconeogenic pathway in strain R30c, we enhanced 3-HP production by 34% (Fig. 6b). Following this, we overexpressed the mutated genes in the glycolytic pathway, including *PYK1m*, *GPM1m2*, and *FAB1m1*, resulting in strain RC04p, which exhibited an additional 31% increase in 3-HP production. Furthermore, the introduction of *gapN* into RC04p led to the creation of strain RC05p, which displayed a further 24% improvement in 3-HP yield compared to strain RC04p. Interestingly, while the overexpression of wild-type *PFK1* together with mutated *PKF2m* in RC05p decreased the 3-HP yield, the same combination in RC04p led to an 8% increase in 3-HP yield compared to strain RC04p, suggesting a potential link between PFK and cofactor metabolism. Subsequently, we co-overexpressed three TFs, namely *GCR1*, *GCR2*, and *MBP1*, and observed a slight increase in 3-HP production. A similar outcome was noted when we downregulated *MPC3* expression using the *pIDP2* promoter, indicating a potential limitation in the downstream pathway. To test this, we introduced an additional copy of *MCRN* and *MCRCm* into strain RC05p, resulting in strain RC05Bp, which exhibited a 29% increase in 3-HP production. Further downregulation of *MPC3* expression in RC05Bp generated strain RC10Bp, which led to an additional 12% increase in 3-HP production. We then introduced the PDH bypass pathway by individually or in combination overexpressing *PDC1*, *ACSm*, *eutE*, *tACC1*, and *ACC1*** in various strains, such as RC05p, RC10p, and RC10Bp. The optimal result was achieved when *ACC1*** was overexpressed in strain RC10Bp, resulting in a 3-HP level of 6.05 g/L, which represents a 47% enhancement compared to RC10Bp. Utilizing the promoter *pARO9* to regulate the expression of *FAS1* in the RC10Bp strain led to an 11% increase in 3-HP levels. To determine whether the overexpression of MCR could drive more carbon flux from the central metabolic pathway to 3-HP production, we inserted the *MCRN* and *MCRCm* genes into a plasmid and employed different promoters to control their expression. Contrary to our expectations, it was specifically the combination of using the promoter

*pMDH2* for *MCRN* and the promoter *pHXT7* for *MCRCm* that yielded a significant boost in 3-HP production, reaching a concentration of 7.46 g/L. To visualize the distribution of metabolic flux in the central carbon metabolic pathway, we performed fermentation (Supplementary Fig. 18) and parsimonious flux balance analysis (pFBA) (Supplementary Fig. 19). The results revealed that a greater proportion of flux was channeled from the central metabolic pathway towards 3-HP synthesis, consistent with the design of our engineered strain for 3-HP production.

To produce flaviolin, which was repurposed as a malonyl-CoA biosensor[57], we deleted the *MCR* genes in strains R30C, RC05p, RC10Bp, and RC32F2p8, and subsequently introduced *rppA*. This resulted in the creation of strains RE01p, RE04p, RE05p, and RE03p. Among these, RE03p exhibited a significant increase in flaviolin titer by 46% and a 137% higher yield per OD compared to RE01p (Supplementary Fig. 21). In contrast, while RE04p and RE05p had a higher yield per biomass, they produced less flaviolin than R30C. Our modifications not only enhanced the conversion efficiency of xylose to acetyl-CoA but also increased the NADPH synthesis flux (Supplementary Fig. 19). Importantly, flaviolin synthesis requires acetyl-CoA but does not necessitate NADPH. Consequently, higher flux of acetyl-CoA in the engineered strain increased flaviolin production, whereas the synthesized NADPH adversely impacted its synthesis level. Therefore, the current engineered strain is more suitable for producing acetyl-CoA-derived products that also require NADPH as a cofactor.

## Discussion

The development of yeast platform strains for acetyl-CoA-derived products has garnered significant attention due to the potential of acetyl-CoA as a precursor for various biochemicals. Moreover, the acetyl-CoA node serves as a bridge between cellular central metabolism and bioproduct generation pathways. Over millions of years of evolution, central carbon metabolism has evolved to prioritize biomass production over the generation of desired products[1]. Consequently, deregulating intracellular central metabolism is a crucial prerequisite to maximize carbon flux toward the final product.

To achieve the desired cellular phenotype, it is often necessary to control the expression of multiple genes, typically through the regulation of TFs. However, unlike *E. coli*, which has a hierarchical TF network that simplifies the process of redirecting carbon fluxes for the overproduction of specific molecules, yeast poses a more complex challenge[1]. Its central carbon metabolism is intricately regulated with a 'flat' structure, involving over 100 TFs, 78% of which are interlinked through cross-regulation in a large internal loop[1]. Due to this complex system of transcriptional regulation, there have been limited successful attempts to alter yeast's cellular phenotype by manipulating the activating or repressive TFs[14,58,59]. Previous studies have shown that modifying TFs can result in increased yeast biomass and ethanol production, although the improvements are typically modest[14,58]. In line with these findings, our study observed only a slight increase in 3-HP

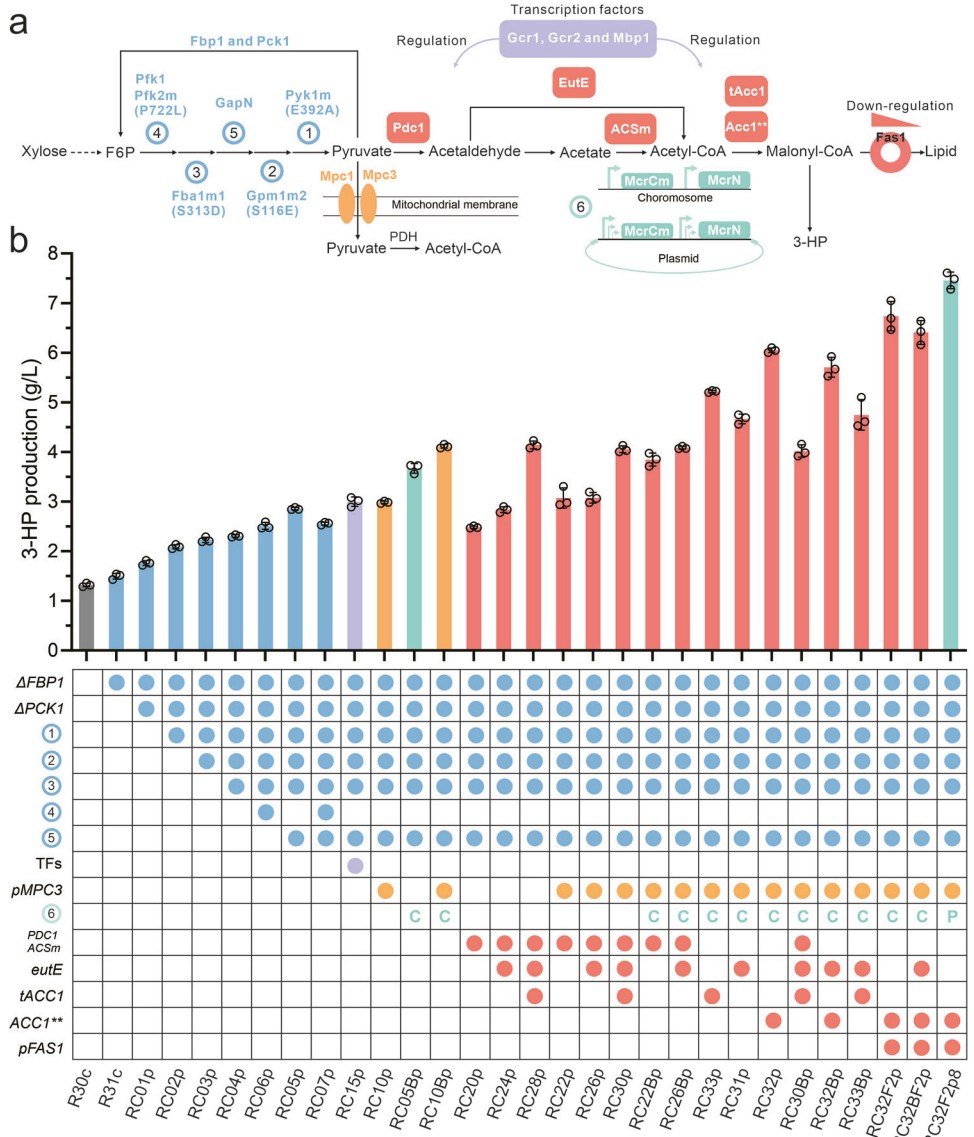

**Fig. 6 | Combinatorial optimization to increase the carbon flux through central metabolic pathway. a** Schematic representation of the key deregulated gene targets. The strategy integrates five components: gluconeogenesis and glycolysis (blue); transcription factors (purple); mitochondrial pyruvate transporter (yellow); PDH bypass (orange); and 3-HP formation (green). Enzymes depicted in these pathways are detailed in Figs. 1, 4, and 5. **b** 3-HP production by combined engineered strains with the indicated genetic modifications from (**a**). C in green, chromosome integration; P in green, plasmid expression. All strains were cultivated in a minimal medium containing 2% xylose as the sole carbon source. R30c was used as the starting strain. All data presented in this figure represent the mean from $n = 3$ biologically independent samples, with error bars indicating the standard deviation. Source data are provided as a Source Data file.

levels after engineering the *GCR1*, *GCR2*, and *MBP1* TFs (Fig. 3a). This may also be attributed to our limited understanding of yeast regulation. To enhance our ability to engineer TFs in yeast, it is necessary to employ additional methods, such as chromatin immunoprecipitation with lambda exonuclease digestion (ChIP-exo) analysis[60], to gain a comprehensive view of the yeast transcriptional landscape.

By examining 3-HP synthesis levels and metabolic flux changes (Supplementary Fig. 19), we pinpointed three key metabolic nodes that are pivotal in the central carbon metabolism when xylose serves as the carbon source (Supplementary Fig. 20). During glycolysis, fructose 6-phosphate emerges as a critical node, steering carbon flux downstream to pyruvate through glycolysis and upstream to glucose 6-phosphate and the pentose phosphate pathway via gluconeogenesis (Supplementary Fig. 9). At this juncture, PFK plays a decisive role in controlling the flux direction (Fig. 4e), which is essential for regulating glycolysis when xylose is the carbon source. This contrasts with prior research indicating that the glycolytic pathway's rate is typically controlled by PYK when glucose is the substrate[35]. Pyruvate faces a decision point: to enter mitochondria or proceed to the PDH bypass. With glucose as the carbon source, the Crabtree effect directs carbon toward the PDH bypass. However, with xylose, the Crabtree effect is diminished, allowing greater carbon flux into the TCA cycle. By managing pyruvate transport and PDH bypass, we can activate the Crabtree effect, thereby augmenting carbon flux to the PDH bypass (Fig. 5a–e). Malonyl-CoA serves as a bifurcation for 3-HP and fatty acid synthesis. By leveraging xylose-responsive promoters and biosensors to modulate the expression of ACC mutants and FAS (Fig. 5f–h), we sucessfully redirected carbon flux toward 3-HP synthesis.

For the commodity chemical 3-HP, metrics such as titer, rate, and yield are crucial. To maximize 3-HP synthesis, various metabolic pathways have been developed using intermediates like glycerol, β-alanine, and malonyl-CoA. Among these, the glycerol pathway offers the shortest route from sugar to 3-HP but requires the costly cofactor vitamin B12. For instance, 50 μM B12 was

needed to achieve 54.8 g/L 3-HP with a carbon yield of 0.49 g/g sugar using glucose and xylose as carbon sources[61]. Additionally, to maintain high production levels, inducers like 0.1 mM IPTG and antibiotics such as 5 μg/mL chloramphenicol and/or 25 μg/mL kanamycin were required, further increasing production costs[61]. In contrast, the β-alanine and malonyl-CoA pathways have garnered more attention due to their avoidance of expensive cofactors. By integrating the β-alanine pathway into yeast strains capable of consuming xylose and using controlled fed-batch cultivation, the engineered strain achieved a 3-HP titer of approximately 7.4 g/L within 120 h, with a product yield of 0.17 g/g xylose[62,63]. By integrating β-alanine pathway genes into *Aspergillus* species and optimizing the cultivation conditions, the conversion rate of 3-HP was increased to 0.24 g/g glucose and xylose, resulting in final concentrations of 36 g/L[64]. The malonyl-CoA pathway also showed promise. Through engineering oleaginous yeast *R. toruloides*, which naturally has a high flux towards malonyl-CoA, the obtained strain synthesized 45.4 g/L of 3-HP with a yield of 0.11 g/g from glucose and xylose mixture[65]. More recently, by enhancing the co-utilization of glucose and xylose in the industrial yeast *Ogataea polymorpha*, the obtained strain produced 5.4 g/L of 3-HP in shake flask and 79.6 g/L in fed-batch fermentation[66], at yields of 0.18 g/g and 0.34 g/g glucose-xylose mixture, respectively. In this work, we systematically engineered the regulation of central carbon metabolism, resulting in a strain capable of producing 7.46 g/L of 3-HP in shake flask, with a significantly improved product yield of 0.37 g/g xylose (Supplementary Data 8). This yield demonstrates an advantage in xylose conversion efficiency over previously reported strains.

In conclusion, our research employed a systematic, modular deregulation strategy on central carbon metabolism, resulting in a significant enhancement of the metabolic conversion of xylose into the acetyl-CoA derivative 3-HP. The final strain, RC32F2p8, exhibited a 4.7-fold improvement compared to the module I-optimized strain R30c, and approximately a 100-fold improvement compared to the initially obtained strain X469p. To drive future advancements, it is necessary to develop a more inclusive and streamlined regulatory framework with a sophisticated strategy that adapts to changes in key metabolite levels, ensuring efficient and direct carbon flux towards acetyl-CoA-derived products. Additional strategies to further increase 3-HP production, such as engineering the efflux mechanisms of 3-HP to increase extracellular secretion, employing laboratory evolution to enhance tolerance to 3-HP, maintaining precise pH control to facilitate extracellular dissociation of 3-HP and minimize its reuptake into the cell, as well as developing rapid isolation methods for 3-HP during the fermentation process.

## Methods

### Plasmids and strains
In this study, all plasmids are listed in Supplementary Data 1. The plasmids were propagated in the *E. coli* strain DH5α. All *S. cerevisiae* strains were derived from the parent strain IMX581[67], which is derived from CEN.PK113-5D (*MATa ura3-52 can1::cas9-natNT2 TRP1 LEU2 HIS3*). This parent strain features an integrated Cas9 expression cassette regulated by the constitutive TEF1 promoter. Detailed information about these strains is available in Supplementary Data 2. All exogenous genes incorporated into the yeast were synthesized by Genscript and codon-optimized for efficient expression in *S. cerevisiae* (Supplementary Data 5).

### Reagents
In this study, all primers were synthesized by either Sigma-Aldrich or Eurofins. DNA-related procedures, including purification, gel extraction, and plasmid isolation, were performed using kits acquired from Thermo Fisher Scientific. For high-fidelity DNA polymerization,

Phusion High-Fidelity DNA Polymerase from New England Biolabs and PrimeStar DNA polymerase from TaKaRa Bio were utilized. For PCR validation, we utilized SapphireAmp Fast PCR Master Mix, sourced from TaKaRa Bio. RNA extraction was carried out using the RNeasy Mini Kit from Qiagen. Additionally, all chemicals employed in the study were procured from Sigma-Aldrich or Merck Millipore.

### Media
In this study, *E. coli* strains were selected and cultivated on Luria-Bertani (LB) agar plates containing 50 mg/L ampicillin. Subsequent liquid cultures were also grown in LB medium, supplemented with the same concentration of ampicillin. Yeast competent cells were grown in YPD medium, which contained 10 g/L yeast extract, 20 g/L peptone, and 20 g/L glucose. For strains harboring URA3-based plasmids, selection was performed using synthetic complete media devoid of uracil (SD-URA). This medium comprised 6.7 g/L yeast nitrogen base without amino acids (YNB), 0.77 g/L complete supplement mixture without uracil (CSM-URA), 20 g/L glucose, and 20 g/L agar powder. To counter-select against the URA3 marker, we used SD + 5'-FOA plates, which contained 6.7 g/L YNB, 0.77 g/L CSM-URA, and 0.8 g/L 5-fluoroorotic acid (5-FOA). Strains equipped with a kanMX cassette were selected on YPD plates, supplemented with 200 mg/L G418, sourced from Formedium. The minimal medium used consisted of 7.5 g/L $(NH_4)_2SO_4$, 14.4 g/L $KH_2PO_4$, and 0.5 g/L $MgSO_4 \cdot 7H_2O$[39]. Either 20 g/L of xylose or 20 g/L of glucose served as the carbon source. This medium was supplemented with vitamin and trace metal solutions, and the pH was adjusted to 6.5 using KOH. When necessary, uracil supplementation was added at a concentration of 60 mg/L.

### Plasmid construction
To construct overexpression plasmids, we employed the Gibson Assembly Master Mix from New England BioLabs, fusing the pSPGM2 vector with amplified gene fragments. For this amplification, we used the genomic DNA of IMX581 as the template for all native elements, including promoters, genes, and terminators. In cases requiring codon-optimized genes, synthetic sequences served as the amplification templates. Guide RNA (gRNA) plasmids were similarly assembled using the Gibson Assembly Master Mix. To identify the optimal 20-bp gRNA sequences targeting specific genes or genomic loci, we cross-referenced potential gRNA candidates against possible off-target sites within the entire CEN.PK113-7D genome. This analysis was facilitated by the CRISPRdirect web tool, accessible at http://crispr.dbcls.jp/. The integrity of all generated plasmids was subsequently confirmed through sequencing.

### Strain construction
We employed CRISPR/Cas9-based techniques for gene integration, deletion, and knockdown, in line with methods previously outlined[67]. Briefly, to overexpress genes, we integrated gene expression cassettes at genomic loci known for high-level and stable heterologous gene expression via the CRISPR/Cas9 system[68]. Components of these cassettes, including homologous arms, promoters, genes, and terminators, were amplified using Phusion high-fidelity DNA polymerase. Subsequently, PrimeSTAR high-fidelity polymerase was utilized to assemble these parts into the complete gene expression cassette, following the overlap extension polymerase chain reaction (OE-PCR) procedure[69]. For gene deletions, we replaced the target gene's open reading frame (ORF) with deletion fragments containing 50-bp upstream sequences of the gene's promoter and 50-bp downstream sequences of its terminator. For promoter-based gene knockdowns, a fragment consisting of an alternative promoter with varying strength is used to replace the original promoter. This fragment is flanked on both sides by 50-bp sequences that are homologous to the respective regions of the gene and its native promoter. During yeast transformation, 500 ng

of DNA fragments were used to repair the chromosomal double-strand break. Transformants were selected on SD-URA plates for 3 days. In an alternative gene deletion strategy, the KanMX marker was employed. Two homologous arms, each ranging between 400 and 600 bp, were amplified from IMX581 genomic DNA. The KanMX gene was similarly amplified using the pUG6 plasmid as a template. These elements were then fused to form the deletion cassette via the above-mentioned OE-PCR method. 1 μg of DNA fragments were used to facilitate the homologous recombination. Transformed colonies were isolated by culturing on YPD agar plates containing 200 mg/L of G418 for 3 days. The colonies were then cultured overnight in YPD medium. Genomic DNA was extracted using a protocol previously described[70]. Specifically, cells were treated with a 0.2 M lithium acetate solution containing 1% SDS and subsequently heated at 75 °C for 15 min. Genomic DNA was precipitated and purified in 70% ethanol, followed by drying and dissolution in 50 μL water. The successful integration or deletion was confirmed through PCR amplification using appropriate primer pairs.

## Growth measurement

Growth measurements were conducted using the Growth Profiler 960 system. Individual colonies were inoculated into 2 mL of minimal medium and incubated with 200 rpm agitation overnight. Following incubation, the preculture was centrifuged at $3000 \times g$ for 5 min to separate the supernatant from the cell pellet. The cell pellet was then resuspended in an uracil-free minimal medium to achieve an initial optical density ($OD_{600}$) of 0.1. Subsequently, 250 μL of the cell suspension was cultivated in 96-well microplates under conditions of 250 rpm agitation and a temperature of 30 °C. $OD_{600}$ were derived from a standard curve generated by correlating green values obtained from the Growth Profiler with $OD_{600}$ readings taken from a spectrophotometer.

## Promoter strength measurement

Promoter sequences (Supplementary Data 4), along with the homologous arms, fluorescent proteins (either GFP or RFP), and terminators, were assembled through OE-PCR. These assembled fragments were integrated into the yeast chromosome at the XI-1 locus using the CRISPR/Cas9 methodology described above. Successfully transformed colonies were cultured overnight in 2 mL of minimal medium, agitated at 200 rpm. Post-incubation, the cultures were centrifuged for 5 min at $3000 \times g$ to pellet the cells, separating them from the supernatant. The resulting cell pellets were resuspended in minimal medium supplemented with either 2% glucose or 2% xylose as the carbon source, and the initial $OD_{600}$ was calibrated to approximately 0.1. One milliliter of this resuspended culture was then transferred to a 48-well FlowerPlate, with a shaking frequency set at 1200 rpm. Fluorescent signals were quantified using excitation/emission wavelengths of 485/515 nm for GFP and 588/633 nm for RFP. All fluorescence readings were background-corrected and normalized to the $OD_{600}$.

## Flask cultivation

Three biological replicates of yeast colonies were pre-cultured in 2 mL of minimal medium and subjected to overnight incubation at 30 °C with agitation at 200 rpm. A designated volume of the cell culture was then centrifuged at $3000 \times g$ for 5 min to pellet the cells. Subsequently, the cells were washed with water through an additional 5-min centrifugation at $3000 \times g$. The washed cells were then transferred to 125 mL shake flasks containing 20 mL of minimal medium, supplemented with 2% glucose or 2% xylose as the carbon source. The initial $OD_{600}$ was adjusted to approximately 0.1. The cultures were then incubated at 30 °C with agitation at 200 rpm. For strains used to calculate the synthesis rate of the product, we collect samples every three hours to measure the 3-HP content.

## Fermentation

Pre-cultures of R30C, RC10Bp, and RC32F2p8 were grown in minimal medium containing 2% glucose and used to inoculate 20 mL of the same medium in 100 mL shake flasks. Bioreactor fermentations for 3-HP production were conducted using the DASGIP® Parallel Bioreactor System (Eppendorf, Germany). Batch fermentations were performed in 0.6 L of minimal medium with 2% xylose. The temperature was maintained at 30 °C, and the pH was controlled at 6.0 through the automatic addition of 2 M KOH. Agitation was adjusted between 400 and 800 rpm to ensure dissolved oxygen levels exceeded 30%, with airflow set to 1 vvm. Batch cultures were initiated by adding all medium components to the reactor, followed by inoculation to achieve an initial $OD_{600}$ of 0.1. $CO_2$ production was monitored by analyzing exhaust gases. The samples were withdrawn every three hours for $OD_{600}$ and metabolites analysis. To investigate the correlation between $OD_{600}$ and dry cell weight (DCW), we diluted the broth to six different $OD_{600}$ values. Then, 5 mL of each dilution was filtered through a pre-weighed 0.45-μm filter membrane (Sartorius Biolab, Göttingen, Germany). The filter was washed once prior to broth filtration and three times afterwards, each time using 5 mL of deionized water. All filters were dried in an incubator at 50 °C until constant weight was achieved before measuring the weight increase. The calculated DCW/$OD_{600}$ ratio was 0.45, 0.64, and 0.66 for strain R30C, RC10Bp, and RC32F2p8, respectively.

## Metabolites measurement

One milliliter of the cultured sample was centrifuged for 3 min at $10,000 \times g$ to separate the cells from the supernatant. The supernatant was then filtered using a 0.2 μm syringe filter. Metabolite concentrations were analyzed using a modified version of a previously established HPLC method[22]. Specifically, an HPLC system from Shimadzu (Japan) equipped with an Aminex HPX-87H column (Bio-Rad, Hercules, USA) was used. The column was maintained at 65 °C, and 0.5 mM $H_2SO_4$ served as the mobile phase, flowing at a rate of 0.4 mL/min over a 30-min analysis period. Concentrations of 3-HP, xylose, xylitol, pyruvate, succinate, glycerol, acetate, and ethanol were detected using an RI-101 Refractive Index Detector. For flaviolin detection, samples were taken in the stationary phase (72 h) and centrifuged at $13,000 \times g$ for 10 minutes at 4 °C. The absorbance of flaviolin was then detected using a Genesys 10UV-VIS spectrophotometer (Fisher Scientific BVBA, Brussel, Belgium) at 340 nm[57].

## Theoretical yield calculation

The maximum theoretical yield of 3-HP was calculated using a previously established method that involves Flux Balance Analysis (FBA)[71]. Specifically, we utilized the genome-scale metabolic model Yeast8 for *S. cerevisiae*[72], into which both the xylose metabolic pathway and the 3-HP production pathway were integrated by adding the respective reactions. Additionally, any genes that were deleted in the experimental strain were represented by the removal of corresponding reactions from the genome-scale model. The maximal secretion rate of 3-HP was then computed by setting a fixed xylose uptake rate of 1 mmol per gram of dry weight per hour. All simulations were conducted in MATLAB using the COBRA Toolbox[73].

## RNA-seq

Yeast cells, grown in minimal medium with glucose, xylose, or ethanol as the carbon source, were harvested at an $OD_{600}$ of approximately 1 and then stored at −80 °C until RNA extraction. The RNeasy Mini Kit was employed for total RNA isolation, and the RNA quality was subsequently verified using the Agilent 2100 Bioanalyzer, following the manufacturer's guidelines. Libraries for RNA sequencing were prepared with the Illumina TruSeq Sample Preparation Kit v2, incorporating a poly-A enrichment step. The prepared samples were clustered using cBot and sequenced as paired end reads (2 × 150 bp) on a HiSeq

2500 platform, in accordance with the manufacturer's instructions. Raw sequencing reads from each sample were aligned to the CEN.PK 113-7D reference genome, available at http://cenpk.tudelft.nl, using TopHat version 2.0.12[74]. Gene expression levels were quantified as FPKM values using Cufflinks version 2.2.166, while raw read counts were determined using the feature Counts module of the Subread package version 1.4.6[75,76]. Differential gene expression analysis was conducted using the DESeq2 method[77].

## TFs analysis

To assess the significance of expression changes in gene sets regulated by TFs, we utilized the Platform for Integrative Analysis of Omics (PIANO) R package[78]. This package calculates the significance of changes in gene expression, taking into account the modulation in the expression levels of genes under the control of specific TFs[79]. Information about these TFs was sourced from the Yeast Search for Transcriptional Regulators and Consensus Tracking Plus database (YEASTRACT+), accessible at http://YEASTRACT-PLUS.org/. Importantly, only TFs with substantiated evidence of DNA binding and expression were considered for analysis in this study. As input, we used the differential expression levels of genes (expressed as Log2 fold changes) along with their corresponding significance levels, which were adjusted using the Benjamini–Hochberg method for multiple comparisons.

## FBA

To perform the flux balance analysis (FBA), we constructed a modified model of the latest yeast GEM[80] by adding the heterologous pathway to 3-HP, GAPN and xylose isomerase. To estimate the flux distribution, we constrained the related exchange rates based on the measurments of 3-HP, xylose, xylitol, pyruvate, succinate, glycerol, acetate, ethanol, and $CO_2$ (Supplementary Data 9). We then performed pFBA to obtain a parsimonious flux distribution and applied 1000 random samplings of the solution space using the Hit and Run algorithm[81]. All simulations were done in MATLAB with the COBRA toolbox[73].

## Statistical analysis

The significance of differences between different groups was evaluated using the two-tailed Student's $t$ test (* represents $p < 0.05$; ** represents $p < 0.01$, and *** represents $p < 0.001$). The statistical analysis of RNA-seq data, utilizing the Benjamini–Hochberg method, was carried out using the R programming language (available at http://www.r-project.org).

## Reporting summary

Further information on research design is available in the Nature Portfolio Reporting Summary linked to this article.

# Data availability

The raw data for RNA-seq have been deposited in the Genome Expression Omnibus (GEO) website (https://www.ncbi.nlm.nih.gov/geo/) under accession code GSE151478. Source data are provided with this paper.

# Code availability

The codes used for the flux balance analysis (FBA) are provided in Supplementary Data 9.

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

## Acknowledgements
We appreciate the helpful discussions with Boyang Ji, Zongjie Dai, Guokun Wang, Quanli Liu, Rui Pereira, and Florian David. We thank Dr. Zongjie Dai for providing PO gene. This work was funded by the Novo Nordisk Foundation (NNF20CC0035580, Y.C.), the Knut and Alice Wallenberg Foundation (J.N.), the Swedish Research Council FORMAS (2022-01130, Y.C.), Carl Tryggers Stiftelse (Y.C.), and Tianjin Synthetic Biotechnology Innovation Capacity Improvement Project (TSBICIP-CXRC-084, X.L.).

## Author contributions
X.L., J.N., and Y.C. conceived the study. X.L. designed the experiments. Y.C. assisted with the experimental design. X.L., Y.W.. X.C., L.E., and C.Z. performed the experiments. X.L., J.N., Y.C., and Y.W. analyzed the data. X.L., Y.C., J.N., and Y.W. wrote the manuscript. All authors read and contributed meaningful discussions for this manuscript.

## Funding

## Competing interests
The authors declare no competing interests.
