## [Transparent Peer Review file · Nature Communications]

Modular deregulation of central carbon metabolism for efficient xylose utilization in *Saccharomyces cerevisiae*

Corresponding Author: Dr Yun Chen

Version 0:

Reviewer comments:

Reviewer #1

(Remarks to the Author)

This manuscript authored by Xiaowei Li et al provides a systematic and modular approach to deregulate the central carbon metabolism in *Saccharomyces cerevisiae* thereby enhanced the conversion of xylose to a product, 3-hydroxypropionic acid (3-HP). The manuscript is overall clearly written, and the study is meticulous and thorough.

Comments:

Lines 110-113 need rephrase. A regulon is a set of gene targets by a TF. The phrase "regulon-controlled...assimilation" in line 112 is confusing.

In lines 121-127, the rationale for choosing 3HP as the product downstream of acetyl-CoA generally makes sense. However, in line 127, was a "lack of regulatory requirements" of this pathway proven previously? Does the strain naturally produce 3HP?

A few places where the description of assays performed were not clear and possibly misleading. E.g., line 174, "reporter TF analysis" does not appear to reflect the work done, which was to identify top up- and down-regulated TF. In Fig 3 legend in line 540, "TF enrichment analysis" was used but I didn't see an "enrichment" aspect in the assay.

I suggest inserting "individual" in line 553 in the fig 4 legend in between 6 and genes to make it deletion of 6 individual genes.

It is not clear which futile cycle the authors refer to in lines 223 and 228 and how the futile cycle is generated. Please clarify.

In lines 240-242, it was stated that the overexpression of all 9 glycolytic genes decreased 3HB. I suggest modifying this sentence by inserting "from expressing either the upper or lower glycolytic enzymes" after 3-HP synthesis in line 241.

Reviewer #2

(Remarks to the Author)

The authors demonstrate a comprehensive engineering approach to enable *S. cerevisiae* to efficiently utilize xylose as a carbon source and produce acetyl-CoA derived 3HP as a product, both of which are not native to the host metabolism. Many interesting findings are presented from engineering the central metabolism to enhance carbon flux from xylose to acetyl-CoA which indeed was a significant shift of metabolism from its native metabolism. However, the overall impact of this study could be questioned due to the remaining inefficiencies in product synthesis, especially in the context of titer, rate and yield. These metrics become even more important for commodity chemical precursors like 3HP as the low product price mandates high productivity, yield and scale for the process to be profitable. Moreover, there is existing literature on 3HP production from glucose and/or xylose at significantly higher metrics (e.g., >60 g/L titer from glucose and xylose mixture using *C. glutamicum* as a host; <https://doi.org/10.1016/j.ymben.2016.11.009>).

The following comments are directed toward the conclusions of the manuscript:

- Authors suggest that the approaches used in this study are not just applicable to the 3HP production but to the general idea

of engineering *S. cerevisiae* for utilization of xylose or even other pentose sugars and production of acetyl-CoA derived products. However, the demonstration of results in this study simply shows the change in 3HP titer but no other byproducts that could be directly affected by each engineering step. It would be helpful to include the change in the key metabolite profile with each modification and provide a valid hypothesis based on the observations.

o To further strengthen the argument that these modifications could enhance flux from xylose to acetyl-CoA, a comprehensive metabolic flux analysis could be useful.

o Showing additional acetyl-CoA derived products and demonstrating universal improvement in multiple acetyl-CoA derived products could make the results more impactful.

- The authors do not discuss the rate and yield of 3HP, both of which are important metrics in metabolic engineering and biomanufacturing. How do these numbers compare with the state-of-the-art processes?

- Which metabolic nodes or enzymes are identified as major bottleneck in the pathway after the optimizations? What measures could be implemented to further optimize the strain to improve 3HP yield?

Reviewer #3

(Remarks to the Author)

This paper fails to raise to the level and bar that would be expected in Nature Communications.

Of most importance, this paper essentially lacks all prior evidence in the literature of similar work and instead tries to pass off this effort as novel.

The selection of yeast promoters for xylose specific traits has already been established in the literature. There is quite a bit of development for this in the current manuscript which clearly ignores all of the past work in the field including DOI: 10.1016/j.jbiosc.2017.08.001 that showcases some similar promoters developed here.

Most curiously, one of the authors here (Nielsen) is a co-author on a similar paper (10.1016/j.meteno.2015.10.001) in which a strain of yeast was engineered for xylose conversion to 3HP. Several other gene targets were identified and this strain was able to achieve 7 g/L production from xylose. This paper is not even cited in this work here.

The division of the various different modules and the regulation here is very unclear and unjustified---is there a biochemical mechanism or rationale or is this more of a conceptual framework? The mitochondrial node makes a divide, but why would regulation need to necessarily be different?

The authors do not fully describe an approach that reaches to the novelty expected in this journal.

Version 1:

Reviewer comments:

Reviewer #2

(Remarks to the Author)

The authors present their main achievement as the systematic engineering of central carbon metabolism regulation to enhance xylose-to-acetyl-CoA flux, thereby increasing 3-HP production. However, their approach of solely reporting 3-HP titers for each genetic modification provides insufficient evidence to support this claim. While the authors have included flux balance analysis (FBA) in the revised version, this computational modeling approach alone cannot provide experimental validation of their hypothesis. To strengthen their argument, I strongly recommend incorporating metabolic flux analysis (MFA), which measures actual intracellular metabolite levels to quantify metabolic fluxes. MFA would not only provide concrete experimental evidence but also offer valuable insights into the metabolic flux shifts accompanying each genetic modification. Furthermore, MFA could provide compelling evidence for the observed metabolic shift to respiratory mode when using xylose as a carbon source. These additional analyses would significantly enhance the paper's impact by providing a deeper understanding of the underlying metabolic mechanisms.

- Table S8 is missing in the supplementary materials.

Version 2:

Reviewer comments:

Reviewer #2

(Remarks to the Author)

I appreciate the authors for sufficiently addressing my comments.

REVIEWER COMMENTS

Reviewer #1 (Remarks to the Author):

This manuscript authored by Xiaowei Li et al provides a systematic and modular approach to deregulate the central carbon metabolism in *Saccharomyces cerevisiae* thereby enhanced the conversion of xylose to a product, 3-hydroxypropionic acid (3-HP). The manuscript is overall clearly written, and the study is meticulous and thorough.

Comments:

Lines 110-113 need rephrase. A regulon is a set of gene targets by a TF. The phrase "regulon-controlled...assimilation" in line 112 is confusing.

Response: Thank you for this comment. We have rephrased the sentence as: "For example, previous research has shown that regulon-controlled expression of galactose catabolic genes supports higher growth rates compared to their constitutive expression".

In lines 121-127, the rationale for choosing 3HP as the product downstream of acetyl-CoA generally makes sense. However, in line 127, was a "lack of regulatory requirements" of this pathway proven previously? Does the strain naturally produce 3HP?

Response: We are sorry for this confusion. The yeast strain does not naturally synthesize 3-HP. Our intention here is to highlight that this pathway is relatively short, requiring only two enzymes, and their catalytic processes are not subject to complex regulation. We acknowledge that the original description was somewhat unclear, so to avoid potential misunderstandings, we have removed the phrase "and its lack of regulatory requirements" (line 125-131).

A few places where the description of assays performed were not clear and possibly misleading. E.g., line 174, "reporter TF analysis" does not appear to reflect the work done, which was to identify top up- and down-regulated TF. In Fig 3 legend in line 540, "TF enrichment analysis" was used but I didn't see an "enrichment" aspect in the assay.

Response: Thank the reviewer for pointing out these issues. We have revised the text to replace "reporter TF analysis" with "differential TF analysis", and we have removed the term "enrichment."

I suggest inserting "individual" in line 553 in the fig 4 legend in between 6 and genes to make it deletion of 6 individual genes.

Response: Thank you for the suggestion. We have added the word "individual" in the Figure 4 legend.

It is not clear which futile cycle the authors refer to in lines 223 and 228 and how the futile cycle is generated. Please clarify.

Response: Thank you for pointing this out. Futile cycling, or substrate cycling, occurs when a metabolic pathway and its reverse pathway are active at the same time. Here, gluconeogenesis, the reverse pathway of glycolysis, appears to be active, as the transcriptome data show expression of enzymes

from both pathways. This suggests the potential for futile cycling. To clarify this, we have revised the sentence to: "These findings suggest that when cells grow on xylose, enzymes from both the gluconeogenic and glycolytic pathways are active and operate in opposite directions, potentially leading to futile cycling".

In lines 240-242, it was stated that the overexpression of all 9 glycolytic genes decreased 3HB. I suggest modifying this sentence by inserting "from expressing either the upper or lower glycolytic enzymes" after 3-HP synthesis in line 241.

Response: Thank you for the suggestion. According to your comment, we changed the sentence to: 'Interestingly, the co-overexpression of all nine genes surprisingly resulted in a decrease in 3-HP synthesis compared to overexpressing either the upper or lower glycolytic enzymes alone, indicating that a balanced interplay between the upstream and downstream glycolysis is crucial'.

Reviewer #2 (Remarks to the Author):

The authors demonstrate a comprehensive engineering approach to enable *S. cerevisiae* to efficiently utilize xylose as a carbon source and produce acetyl-CoA derived 3HP as a product, both of which are not native to the host metabolism. Many interesting findings are presented from engineering the central metabolism to enhance carbon flux from xylose to acetyl-CoA which indeed was a significant shift of metabolism from its native metabolism.

Response: Thank you very much for your valuable comments.

However, the overall impact of this study could be questioned due to the remaining inefficiencies in product synthesis, especially in the context of titer, rate and yield. These metrics become even more important for commodity chemical precursors like 3HP as the low product price mandates high productivity, yield and scale for the process to be profitable. Moreover, there is existing literature on 3HP production from glucose and/or xylose at significantly higher metrics (e.g., >60 g/L titer from glucose and xylose mixture using *C. glutamicum* as a host; <https://doi.org/10.1016/j.ymben.2016.11.009>).

Response: Thank you for your suggestion. We fully agree with the reviewer's perspective that for commodity chemical precursors like 3HP, key metrics such as titer, rate, and yield are indeed particularly important to make the process viable. While we acknowledge that 3-HP production has been extensively studied using glucose and/or xylose as carbon source, prior studies primarily focused on developing and optimizing various 3-HP synthesis pathways (e.g., **glycerol, β -alanine, and malonyl-CoA**), and improving xylose uptake. In our study, the main objective is to systematically deregulate the central metabolic pathways for efficient conversion of xylose into acetyl-CoA derived products, i.e., 3-HP.

In the study referenced (doi.org/10.1016/j.ymben.2016.11.009), employing the glycerol pathway allowed the production of 3-HP to reach 62.6 g/L using glucose, with a yield of **0.51 g/g glucose**. Additionally, when using a mixture of glucose and xylose, the production of 3-HP reached 54.8 g/L, with a carbon yield of **0.49 g/g sugar**. It is not clear how was the performance on xylose alone. Furthermore, to support such high production levels, 50 μ M of vitamin B12 was supplemented, in addition to 0.1 mM isopropyl-beta-D-thiogalactoside (IPTG) to induce high-level expression of genes, and 5 μ g/ml chloramphenicol and/or 25 μ g/ml kanamycin to prevent plasmid loss. Therefore, while the glycerol pathway is relatively straightforward and can achieve high production levels, it also necessitates the addition of expensive cofactors like B12 and inducers, significantly increasing the costs associated with commercial product production.

By introducing the β -alanine pathway into a yeast strain capable of consuming xylose and using controlled fed-batch cultivation, 3-HP production achieved at a titer of approximately 7.4 g/L within 120 hours, with a product yield of **0.17 g/g xylose** (doi.org/10.1016/j.meteno.2015.10.001). Similarly, integrating β -alanine pathway genes into *Aspergillus* species and optimizing the cultivation conditions improved the conversion rate to **0.24 g /g from glucose and xylose**, resulting in a final 3-HP concentration of 36 g/L (doi.org/10.1186/s13068-023-02288-1).

The malonyl-CoA pathway has also been widely applied in various host species. For instance, by integrating multiple engineering efforts and

optimizing the medium in a fed-batch fermentation, the oleaginous yeast *R. toruloides*, produced 45.4 g/L of 3-HP, with a product yield of **0.11 g/g from glucose and xylose** mixture (doi.org/10.1016/j.ymben.2023.05.001). More recently, efforts to enhance the co-utilization of glucose and xylose in the industrial yeast *Ogataea polymorpha*, resulted in production of 5.4 g/L of 3-HP in shake flask batch fermentation and 79.6 g/L in fed-batch fermentation after 192 hours of cultivation, at yields of **0.18 g/g and 0.34 g/g glucose-xylose mixture**, respectively (doi.org/10.1038/s41589-023-01402-6).

In this work, we systematically engineered the regulation of central carbon metabolism, resulting in a strain capable of producing 7.46 g/L of 3-HP in shake flask batch fermentation using 20 g/L of xylose, with a significantly improved product yield of **0.37 g/g xylose**. This yield demonstrates an advantages in xylose conversion efficiency over previously reported strains. We have now incorporated these comparisons in the discussion section for clarity (line 503-525).

The following comments are directed toward the conclusions of the manuscript:

- Authors suggest that the approaches used in this study are not just applicable to the 3HP production but to the general idea of engineering *S. cerevisiae* for utilization of xylose or even other pentose sugars and production of acetyl-CoA derived products. However, the demonstration of results in this study simply shows the change in 3HP titer but no other byproducts that could be directly affected by each engineering step. It would be helpful to include the change in the key metabolite profile with each modification and provide a valid hypothesis based on the observations.

Response: Thank you for your suggestion. We have examined the byproducts xylitol, glycerol, acetate, and ethanol in strains R30C, RC01p, RC05p, RC10Bp, RC32p, and RC32F2p8. Intriguingly, we did not observe any accumulation of these byproducts in these strains. This finding aligns with previous reports, which stated that "no significant amounts of by-products such as acetate, ethanol, or glycerol were detected at the end of the fermentation" (doi.org/10.1016/j.ymben.2014.10.003). A plausible explanation for this phenomenon is that when yeast uses xylose as a carbon source, its metabolic state shifts towards a respiratory mode, unlike the fermentative metabolism typically seen with glucose. Consequently, a portion of the carbon source is released as carbon dioxide rather than being directed towards the formation of byproducts. Additionally, this observation is consistent with our previous findings on the cellular mechanism for xylose utilization in engineered yeast, where yeast metabolism when utilizing xylose leans more towards aerobic glycolysis (doi.org/10.1038/s41929-021-00670-6).

o To further strengthen the argument that these modifications could enhance flux from xylose to acetyl-CoA, a comprehensive metabolic flux analysis could be useful.

Response: According to your suggestion, we have performed the FBA in three representative strains: R30C, RC10Bp, and RC32F2p8, which represent low-, medium-, and high-level 3-HP producers, respectively (see new Supplementary Fig. 18). By comparing the changes in metabolic flux between the RC32F2p8 and R30C strains, we can draw the following conclusions: (1) The flux of fructose-6-phosphate towards glyceraldehyde-3-phosphate and further to pyruvate has been enhanced, while the flux of fructose-6-phosphate flowing back to glucose-6-phosphate through gluconeogenesis has decreased. This indicates that we have achieved a significant increase in

glycolytic flux by regulating the glycolysis module. (2) The conversion of glyceraldehyde-3-phosphate to 1,3-bisphosphoglycerate has switched from synthesizing NADH to exclusively synthesizing NADPH, and the flux of glucose-6-phosphate towards the oxidative pentose phosphate (OPP) pathway has decreased to zero. Since this OPP pathway involves the release of carbon dioxide during NADPH production, our strategy of introducing NADP⁺-dependent glyceraldehyde 3-phosphate dehydrogenase (GapN) not only increases NADPH synthesis but also alters the pathway for obtaining NADPH, reducing carbon loss and subsequently enhancing carbon recovery. (3) The flux of carbon from pyruvate towards mitochondria has decreased, indicating that our control over the intensity of pyruvate transporter proteins has reduced the carbon loss in the TCA cycle, which is part of the respiratory metabolism. (4) The flux of pyruvate towards the PDH bypass pathway has increased significantly, suggesting that our strategy to strengthen this module further promotes the flux of carbon sources towards product formation. In summary, by deregulating central carbon metabolism, we have significantly increased the conversion capacity of xylose to 3-HP through central pathways.

o Showing additional acetyl-CoA derived products and demonstrating universal improvement in multiple acetyl-CoA derived products could make the results more impactful.

Response: Thank you for your suggestion. We deleted the MCR genes in strains R30C, RC05p, RC10Bp, and RC32F2p8 and introduced RppA, a type III polyketide synthase, to produce red-colored flaviolin, which repurposed as an malonyl-CoA biosensor. This led to the generation of strains RE01p, RE04p, RE05p, and RE03p, respectively. When compared to strain RE01p, strain RE03p exhibited increased flaviolin production (titer increased by 46%, and the yield per OD of biomass increased by 137%). Conversely, strains RE04p and RE05p produced less flaviolin than R30C, despite having a higher flaviolin value per biomass (Supplementary Fig. 19). This result indicates that by regulating and modifying the central pathway, we have enhanced the conversion efficiency of sugar to acetyl-CoA. Simultaneously, this modification was accompanied by a switch in the NADPH synthesis pathway, leading to an increase in its synthesis level. For flaviolin, the synthesis of one molecule of flaviolin requires five molecules of acetyl-CoA but does not necessitate the participation of NADPH. Consequently, the accumulation of acetyl-CoA in the engineered strain promotes flaviolin synthesis, whereas the synthesized NADPH adversely affects its synthesis level. In summary, the current engineered strain is more suitable for producing acetyl-CoA-derived products that also require the cofactor NADPH for synthesis. These additional results are presented in the revised results section (line 454-464).

- The authors do not discuss the rate and yield of 3HP, both of which are important metrics in metabolic engineering and biomanufacturing. How do these numbers compare with the state-of-the-art processes?

Response: Thanks for your suggestion. We have provided data on both yield and synthesis rate of strains R30C, RC01p, RC05p, RC10Bp, RC32p, and RC32F2p8 in Table S8. Since there are numerous reported yields but relatively fewer synthesis rates, the revised discussion section primarily focuses on comparing the yields when xylose is used as the carbon source or when a combination of glucose and xylose serves as the carbon source (line 503-525).

- Which metabolic nodes or enzymes are identified as major bottleneck in the

pathway after the optimizations? What measures could be implemented to further optimize the strain to improve 3HP yield?

Response: By analyzing the synthesis levels of 3-HP and changes in metabolic flux, we identified three crucial metabolic nodes in the central metabolic pathway. These nodes include (1) fructose 6-phosphate, (2) pyruvate, and (3) malonyl-CoA, which collectively regulate the branching and direction of carbon flow in central metabolism.

(1) In the glycolysis module, fructose 6-phosphate occupies a pivotal position, functioning as a crucial node that can flow downstream to glyceraldehyde 3-phosphate and further proceed through glycolysis to pyruvate, as well as upstream through gluconeogenesis to glucose 6-phosphate and subsequently into the pentose phosphate pathway. At this critical node, we identify phosphofructokinase (PFK) as a key player in controlling the flow direction (Fig. 4e). It thus serves as a vital control point for regulating glycolysis when xylose is utilized as the carbon source. (2) Pyruvate lies at a critical juncture, deciding whether to enter mitochondria or proceed towards the pyruvate branch pathway. When glucose serves as the carbon source, due to the Crabtree effect, a significant portion of the carbon is directed towards the pyruvate-dehydrogenase (PDH) bypass pathway. Conversely, when xylose is the carbon source, the Crabtree effect is less pronounced, resulting in a larger proportion of the carbon flowing into the TCA cycle and being oxidized. Therefore, by controlling the transport and regulating the PDH bypass, the Crabtree effect can be re-established, enhancing the flow of carbon towards the PDH bypass (Fig. 5a-e). (3) Malonyl-CoA serves as a pivotal branching point for the synthesis of both the product 3-HP and fatty acids, and both the production of malonyl-CoA and its conversion into fatty acids are tightly regulated. By utilizing xylose-responsive promoters to control the expression of ACC mutants and introducing biosensors to modulate the intensity of FAS (Fig. 5f-h), we are able to reconstruct this regulatory node, thereby redirecting a greater proportion of carbon flux towards 3-HP synthesis. We discussed these three key nodes in the revised discussion section (line 487-502).

This work focuses primarily on engineering highly regulated central metabolic pathways and does not delve deeply into other factors that may affect the product yield. However, various other processes can also influence the synthesis levels of 3-HP. Potential strategies include engineering the efflux mechanisms of 3-HP to enhance extracellular secretion, utilizing laboratory evolution to improve 3-HP tolerance, implementing precise pH control to facilitate extracellular dissociation and reduce its reuptake, and establishing methods for rapid isolation of 3-HP during fermentation. We have added these aspects in the last paragraph of the discussion section (line 533-537).

Reviewer #3 (Remarks to the Author):

This paper fails to raise to the level and bar that would be expected in Nature Communications.

Of most importance, this paper essentially lacks all prior evidence in the literature of similar work and instead tries to pass off this effort as novel.

Response: Thank you for this comment. This study builds upon our previous work on xylose metabolism (doi.org/10.1038/s41929-021-00670-6), aiming to increase xylose-to-product conversion by deregulating central carbon metabolism. While we acknowledge that 3-HP production has been extensively studied using glucose and/or xylose as carbon source, prior studies primarily focused on developing and optimizing various 3-HP synthesis pathways (e.g., glycerol, β -alanine, and malonyl-CoA), and improving xylose uptake. We have now incorporated this background information and highlighted how our work differs (line 57-61).

Although some individual targets being explored in earlier studies, there has been limited attention to systematic evaluation of cellular metabolic regulation's impact on metabolic flux, particularly with xylose. The key strength of this work lies in the systematic engineering regulatory aspects of the central carbon metabolism during xylose utilization, integrating approaches at both the transcriptional and protein levels. This was done based on employing five different strategies: the use of xylose-responsive promoters, transcription factors (TFs), heterologous proteins, mutated proteins, and biosensors. The combination of these five different strategies significantly enhanced the conversion efficiency of xylose flux through the central metabolic pathway towards acetyl-CoA-derived products, demonstrating an advantage in xylose conversion efficiency over previously reported 3-HP strains that used xylose or glucose-xylose, as discussed in the revised discussion section (line 503-525).

After systematic engineering, we also summarized three crucial metabolic nodes that play key roles in the central carbon metabolism when xylose is used as the carbon source. In glycolysis, fructose 6-phosphate is a key node that directs carbon flow downstream to pyruvate via glycolysis or upstream to glucose 6-phosphate and the pentose phosphate pathway via gluconeogenesis. At this juncture, phosphofructokinase (PFK) controls the direction of carbon flux (Fig. 4e), making it crucial for regulating glycolysis with xylose as the carbon source. This finding contrasts with prior research that identified pyruvate kinase (PYK) as the primary control point for glycolytic flux when glucose is the substrate (doi.org/10.1099/00221287-147-2-391). Pyruvate serves as another decision point, determining whether carbon enters the mitochondria or proceeds to the pyruvate-dehydrogenase (PDH) bypass pathway. With glucose as the carbon source, the Crabtree effect predominates, directing carbon toward the PDH bypass. However, with xylose, the weaker Crabtree effect allows a greater proportion of carbon to flow into the TCA cycle. By regulating pyruvate transport, we can restore the Crabtree effect, thereby enhancing carbon flux to the PDH bypass (Fig. 5a-e). Malonyl-CoA functions as a key branching point for 3-HP and fatty acid synthesis. Leveraging xylose-responsive promoters and biosensors to control ACC mutants and FAS activity (Fig. 5f-h), we successfully redirect carbon flux toward 3-HP synthesis. These findings are further elaborated in the revised discussion section (line 487-502).

By comparing the differences between our work and previous efforts, and

summarizing our key findings, we hope that we have clearly shown the innovation and relevance of this work in the revised version.

The selection of yeast promoters for xylose specific traits has already been established in the literature. There is quite a bit of development for this in the current manuscript which clearly ignores all of the past work in the field including DOI: 10.1016/j.jbiosc.2017.08.001 that showcases some similar promoters developed here.

Response: Thanks for pointing this out. In the previous work (10.1016/j.jbiosc.2017.08.001), the authors evaluated 30 promoters, primarily selected from central metabolic pathways, most of which were constitutive. Their analysis revealed that these constitutive promoters could be used to control the expression of xylose metabolic genes. In this work, we characterized promoters from three distinct groups: (1) xylose-responsive promoters; (2) glucose-responsive promoters; and (3) constitutive promoters. Our findings demonstrated that employing xylose-responsive promoters to regulate the expression of catabolic genes significantly enhanced xylose utilization efficiency compared to constitutive promoters (Fig. 2b). Additionally, we established a set of xylose-responsive promoters with varying expression levels to fine-tune the expression of different genes in the central pathway.

We apologize for not citing or comparing these differences in our original manuscript. In the revised version, we have included the relevant information to address this oversight (line 101-105).

Most curiously, one of the authors here (Nielsen) is a co-author on a similar paper (10.1016/j.meteno.2015.10.001) in which a strain of yeast was engineered for xylose conversion to 3HP. Several other gene targets were identified and this strain was able to achieve 7 g/L production from xylose. This paper is not even cited in this work here.

Response: Thank you for pointing out this shortcoming. In the paper you mentioned (doi.org/10.1016/j.meteno.2015.10.001), 3-HP production pathway via malonyl-CoA or β -alanine was introduced into a yeast strain capable of consuming xylose. While the malonyl-CoA pathway showed very poor 3-HP production, likely due to lacking overflow metabolism on xylose, the β -alanine pathway displayed a much higher yield on xylose comparing with glucose as the carbon source. Subsequently, through controlled fed-batch cultivation in a fermenter, the engineered strain achieved a 3-HP titer of approximately 7.4 g/L, with a product yield of 0.17 g/g xylose. As mentioned above, the paper that you referred primarily focused on developing and optimizing 3-HP synthesis pathway (β -alanine pathway) using xylose as the carbon source. In this study, by optimizing the central metabolism, we enhanced the malonyl-CoA pathway to achieve a higher 3-HP titer of about 7.5 g/L in shake flasks, along with significantly improved product yield of 0.37 g/g xylose. In the revised discussion section, we have compared our findings with the work you referred and also reviewed other relevant studies that utilized xylose or glucose-xylose as carbon sources (line 503-525).

The division of the various different modules and the regulation here is very unclear and unjustified---is there a biochemical mechanism or rationale or is this more of a conceptual framework? The mitochondrial node makes a divide, but why would regulation need to necessarily be different?

Response: We are sorry that we did not present this more clearly in the original version of the manuscript. We will provide a detailed explanation on

the division of modules and regulation as follows.

Initially, we aimed to regulate the entire central metabolism (targeting the state where glucose-driven metabolism is the desired outcome) using transcription factors (TFs) to optimize glucose-driven metabolism and efficiently direct carbon flow from xylose towards the production of 3-HP. However, due to the complexity of central metabolism and its regulation, we found this approach challenging, and instead divided the central metabolism into different modules for targeted regulation and modification.

In this work, we divided central carbon metabolism into three modules: glycolysis, mitochondrial metabolism, and the pyruvate dehydrogenase (PDH) bypass. The primary rationale for this classification stems from two crucial regulatory mechanisms that are present under glucose metabolism but are diminished under xylose metabolism. When glucose serves as the carbon source, the regulation of central metabolism is relatively clear: (1) Due to the presence of the carbon catabolite repression (CCR) response, glycolysis is enhanced, increasing the flux of glucose towards pyruvate; (2) Because of the Crabtree effect, a significant amount of carbon flux is directed from pyruvate into the PDH bypass pathway rather than entering the mitochondria for respiration. Overall, the existence of these two metabolic mechanisms directs a large portion of glucose towards acetyl-CoA synthesis. In contrast, when xylose is used as the carbon source, the attenuation of these two regulatory mechanisms results in a more dispersed carbon flux, making it difficult to efficiently channel carbon towards acetyl-CoA synthesis. Furthermore, since these mechanisms primarily involve glycolysis, mitochondrial respiration, and PDH bypass metabolism, we have accordingly categorized the central carbon metabolism into these three modules. We supplemented this information in the revised manuscript (line 201-207, 212-213, 316-318, and 344-345).

The main reasons for regulating mitochondrial metabolism are as follows: (1) unlike glucose, xylose does not trigger a strong Crabtree effect, resulting in a substantial amount of pyruvate entering the mitochondria for oxidation via the TCA cycle and respiratory metabolism. To address this, it is necessary to reduce pyruvate flux into the mitochondria by modulating its transport; (2) In *S. cerevisiae*, the mitochondrial pyruvate carrier (Mpc) is encoded by the genes *MPC1*, *MPC2*, and *MPC3*, which form two transporter complexes: MPC_{FERM} (Mpc1-Mpc2) and MPC_{OX} (Mpc1-Mpc3). To reduce Mpc transport capacity, one would target the expression of these genes. Decreasing the expression of *MPC1* would lower Mpc transport under both glucose and xylose condition, whereas reducing MPC2 expression affects only under glucose conditions. Therefore, decreasing the expression of MPC3 would specifically reduce pyruvate transport into mitochondria under xylose conditions. Overall, in this study we designed rational interventions to regulate pyruvate transport, primarily based on the distinct metabolic responses of pyruvate when glucose or xylose is used as the carbon source.

The authors do not fully describe an approach that reaches to the novelty expected in this journal.

Response: Thank you for your valuable comments. we are sorry for the unclarity. We have revised our manuscript according to your suggestions. As mentioned above, we have made several changes in the revised manuscript to better highlight the innovations of this work: (1) the revised introduction (line 57-61) emphasizes the key innovations of this work; (2) the main findings are summarized in the updated discussion section (line 487-502); (3) a comparison of our results with previously published studies is provided in the

revised discussion (line 503-525). We, therefore, hope that we have clearly shown the innovation and relevance of this work in the revised version.

REVIEWER COMMENTS

Reviewer #2 (Remarks to the Author):

This manuscript authored by Xiaowei Li et al provides a systematic and modular approach to deregulate the central carbon metabolism in *Saccharomyces cerevisiae* thereby enhanced the conversion of xylose to a product, 3-hydroxypropionic acid (3-HP). The manuscript is overall clearly written, and the study is meticulous and thorough.

Comments:

The authors present their main achievement as the systematic engineering of central carbon metabolism regulation to enhance xylose-to-acetyl-CoA flux, thereby increasing 3-HP production. However, their approach of solely reporting 3-HP titers for each genetic modification provides insufficient evidence to support this claim. While the authors have included flux balance analysis (FBA) in the revised version, this computational modeling approach alone cannot provide experimental validation of their hypothesis. To strengthen their argument, I strongly recommend incorporating metabolic flux analysis (MFA), which measures actual intracellular metabolite levels to quantify metabolic fluxes. MFA would not only provide concrete experimental evidence but also offer valuable insights into the metabolic flux shifts accompanying each genetic modification. Furthermore, MFA could provide compelling evidence for the observed metabolic shift to respiratory mode when using xylose as a carbon source. These additional analyses would significantly enhance the paper's impact by providing a deeper understanding of the underlying metabolic mechanisms.

Response : We sincerely thank the reviewer for their insightful comments and suggestions. We acknowledge the importance of metabolic flux analysis (MFA) in studying metabolic pathways, but we would like to clarify the relationship between metabolite concentrations and fluxes. While metabolite measurements offer valuable insights, it is important to note that fluxes can be highly sensitive to changes in metabolite concentrations. In the fields of enzyme kinetics and metabolic engineering, it is widely accepted that most enzymes in microorganisms, including many central metabolic enzymes, exhibit K_m values in the millimolar range, whereas intracellular metabolite concentrations typically lie within the micromolar to low millimolar range (doi.org/10.1038/nchembio.186). This implies that the K_m values of a large number of enzymes are on the same order of magnitude as metabolite concentrations, meaning that even minor variations in metabolite levels can lead to significant flux changes (doi.org/10.1093/jxb/eri011). Moreover, fluxes are influenced not only by metabolite concentrations but also by enzyme levels (doi.org/10.1038/s44320-025-00090-9), making it challenging to infer flux distributions solely from metabolite data.

Regarding MFA, we assume the reviewer is referring to flux analysis based on ^{13}C -labeled substrate feeding. While historically a widely used method, recent advancements in enzyme-constrained metabolic models have demonstrated that computational approaches can provide equally reliable flux estimates (doi.org/10.1016/j.ymben.2022.09.002). MFA requires extensive experimental data, and its accuracy is highly dependent on measurement precision. In contrast, FBA offers a computationally efficient and scalable approach that integrates genome-scale metabolic models to predict flux distributions, though its accuracy relies on model quality and precise carbon exchange data.

In our study, we initially employed FBA using the most established metabolic model available (doi.org/10.1038/s44320-024-00060-7). However, we acknowledge that data from shake flask fermentations were limited by the lack of exhaust gas monitoring, which prevented us from tracking carbon dioxide release. To address this and enhance the robustness of our results, we performed controlled fermentations with three strains—R30C, RC10Bp, and RC32F2p8 (representing low, medium, and high 3-HP production levels, respectively). In these experiments, we precisely controlled pH and dissolved oxygen levels while monitoring CO₂ emissions (Supplementary Fig. 18). These controlled conditions enabled us to calculate carbon source recovery, achieving a recovery rate of 97.3% for R30C, 97.9% for RC10Bp, and 95.6% for RC32F2p8, thereby enhancing the reliability to our metabolic flux analysis.

Supplementary Fig. 18 Fermentation of engineered strains.

The engineered low-level 3-HP producer, R30C (left), the medium-level 3-HP producer, RC10Bp (middle), and the high-level 3-HP producer, RC32F2p8 (right), were cultivated in a fermentor. Specifically, all strains were grown in a minimal medium supplemented with 2% xylose, at 30° C and pH 6.0.

Using a refined parsed FBA (pFBA) approach, we confirmed that the flux from xylose to pyruvate, and subsequently to 3-HP, was significantly enhanced in RC32F2p8, aligning with our prior findings on central carbon metabolism (Supplementary Fig. 19). To further investigate key metabolic nodes in xylose metabolism, we performed random sampling approach to estimate potential flux distributions (Supplementary Fig. 20). The results highlighted significant flux differences at three critical metabolic nodes, especially between the strains R30C and RC32F2p8, emphasizing their pivotal role in shaping overall metabolic flux patterns.

Supplementary Fig. 19 Calculated metabolic flux distributions via parsimonious flux balance analysis (pFBA).

Strains R30C (left), RC10Bp (middle), and RC32F2p8 (right) were used for pFBA. Based on pFBA, the fluxes of various metabolites were calculated relative to the uptake of 100 mmol of xylose. All strains were grown in fermentors containing a minimal medium supplemented with 2% xylose.

Supplementary Fig. 20 The distribution of calculated flux at three important nodes.

Figures **a-c** show the distribution of calculated flux at the F6P, Pyr, and Mal-CoA nodes, respectively. Each reaction's flow rate was randomly calculated 1000 times. The three strains were grown in fermentors containing a minimal medium supplemented with 2% xylose. G3P, Glyceraldehyde 3-phosphate; S7P, Sedoheptulose 7-phosphate; X5P, Xylulose 5-phosphate; E4P, Erythrose 4-phosphate; F6P, Fructose 6-phosphate; F1,6P, Fructose 1,6-bisphosphate; G6P, Glucose 6-phosphate; PEP, Phosphoenolpyruvate; Pyr, Pyruvate; OAA, Oxaloacetate; Ac-CoA, Acetyl-CoA; Mal-CoA, Malonyl-CoA; FAA, Fatty Acid.

These controlled fermentation-based analyses strongly support our conclusion that deregulation of central carbon metabolism improves the conversion of xylose to 3-HP via key central metabolic pathways. We hope the reviewer finds these improvements compelling and recognizes the validity of our conclusions based on this enhanced methodology.

- Table S8 is missing in the supplementary materials.

Response : We apologize for the omission of Table S8 in the previous version of the supplementary materials and appreciate you bringing this to our attention. The table has been included in the current revised version.

REVIEWERS' COMMENTS

Reviewer #2 (Remarks to the Author):

I appreciate the authors for sufficiently addressing my comments.

Response: We deeply appreciate the reviewer's acknowledgement of our supplementary experiments.